# Voices from the margins: How national stories are linked with support for populist radical right parties

Odelia Oshri[1]*, Eran Amsalem[2], Shaul R. Shenhav[1]

**1** Department of Political Science, Hebrew University of Jerusalem, Jerusalem, Israel, **2** Department of Communication, Hebrew University of Jerusalem, Jerusalem, Israel

* Odelia.oshri@mail.huji.ac.il

## Abstract

How do national stories shape voter behavior? Do they affect all voters equally, or are some groups more influenced by these narratives? This article examines the impact of "boundary national stories," which highlight clear distinctions between "us" and "them" in national identity, on voting patterns for populist radical right parties (PRRPs). Using original representative election surveys conducted in four Western democracies, we find that voters who embrace a Boundary national story are more likely to vote for Populist Radical Right Parties (PRRPs) than those who do not hold such stories, and that the electoral effect of such stories is more salient for marginalized groups in society. Our findings demonstrate that, while national stories can foster cohesion, they can also drive us apart and polarize our politics. We conclude by discussing the broader implications of these findings for the study of populism in political science.

**Data Availability Statement:** Data and code and available at https://dataverse.harvard.edu/dataset.xhtml?persistentId=doi:10.7910/DVN/HUIFRV.

**Funding:** This research was supported by the Israel Science Foundation (Grant No. 1400/14) and by

## 1. Introduction

Populist radical right parties (PRRPs) have become a significant force in politics over the last few decades, joining and even leading governments in a growing number of countries [1–3]. This group of parties adhere to an ideology characterized by authoritarianism, nativism, and populism [4] and take radical positions on cultural issues with nativism, a combination of nationalism and xenophobia, being central to their ideology. They are regarded as populist, as they profess to represent the will of the people, as opposed to that of a putative "other," commonly depicted as a corrupt and self-serving elite [5]. Such parties have significantly impacted policy-making, and in certain cases, they have eroded democratic norms and institutions. Despite an increase in research into mass support for PRRPs, there is still no consensus on what drives this tendency, but the main impetus is commonly ascribed to cultural [3, 6, 7] and economic factors [8, 9]. This paper argues that the explanation for the emergence and success of right-wing populism lies not only in the cultural factors or the economic ones, but at the intersection of both, with the perception of the nation serving as an important cultural factor. The paper thus joins a growing group of scholars who account for the support of the radical right based on a combination of economic and cultural factors [5, 10–12].

the Thyssen Foundation Grant Ref. 20.17.0.047PO. The Danish survey was funded by Danish Council for Independent Research (FSE) (Grant No. 0602-02106B). The funders had no role in study design, data collection and analysis, decision to publish, or preparation of the manuscript.

**Competing interests:** The authors have declared that no competing interests exist.

We suggest that a good predictor of people's vote choice is their perception of their nation, as indicated by the types of stories they tell about their political community, its origin, history, and anticipated future prospects. In order to understand and measure people's perception of their nation, we argue that the political landscape in advanced democracies is divided not simply between those with stronger or weaker national attachments, but rather between groups that adhere to fundamentally different types of nationalism [13–15]. Accordingly, we propose a categorization of nationalism based on distinct ways in which people relate to the national community and define its boundaries. Through this framework, we aim to demonstrate the influence of various types of national stories on partisan identities, and ultimately, voter choice.

National identity, represented here as the national story, can embody shared norms and goals within the group, but it can also be defined through "relational comparisons" with outgroups [16]. Therefore, to examine the content and meaning individuals attribute to national identity, we have developed an analytical framework that encompasses three types of stories, corresponding to broad political divisions in advanced democracies: *Survival*, *Self-expression*, and "us versus them" *Boundary* stories. While the first two kinds encapsulate the norms, goals, and values of the national group, the *Boundary* story defines the national community relationally and negatively, in terms of exclusion.

Two questions are addressed in the paper: (a) What characterizes national stories held by populist radical right voters? and (b) What kind of voters are more affected by these stories? Our investigation is based on original surveys in four countries: Denmark, the Netherlands, the United Kingdom, and the United States. We find that the national stories embraced by PRRP voters differ significantly from those of mainstream party supporters. Specifically, individuals who embrace national stories in the *Boundary* category, emphasizing conflict between a political community and other groups, are more likely to vote for PRRPs. This finding is consistent across all four cases investigated, with a more pronounced effect among individuals positioned at the bottom of a society's power structure.

Our findings contribute directly to the growing empirical literature that locates the popularity of PRRPs on the interface of economic and cultural factors. We empirically demonstrate how a motley slew of grievances find voice in a unifying *Boundary* story type that affords stigmatized groups a positive distinctiveness.

The rest of the paper is divided to five sections. In Section 2 we situate our argument within the literature on the support bases of populist parties. Section 3 outlines the contribution of narrative approaches to the study of politics and, relying on research of political competition, offers an initial typology of national stories. Section 4 presents and elaborates the paper's main argument, making a case for a synthesis of cultural and economic explanations for the support of right-wing populism. Section 5 describes the estimation strategy employed and the data analyzed. Section 6 presents our results and Section 7 is a conclusion.

## 2. Cultural and economic approaches to the study of populist radical right parties

The extensive scholarship on the demand side of right-wing populism [17, 18] is divided with respect to explanations for the success of this party family, with one side emphasizing economic, while the other, cultural factors [2]. According to the economic argument, globalization, open borders, and technological developments have made life less secure for manual workers and the rural population while privileging already highly educated urban dwellers, and as a consequence, has created deep divisions among citizens [19, 20]. It is not surprising,

therefore, that support for PRRPs is the strongest among people facing economic hardship, those who have lost out to globalization—the "left-behinders."

Indeed, researchers of populism have amply shown that PRRP voters have distinctive socio-structural attributes. In particular, PRRPs garner support mainly among the manual working class [21] and individuals with low-to-intermediate levels of education [22], both of which are overrepresented in manual and routine jobs—the type of jobs most threatened by globalization and competition with immigrants. Another salient factor is the urban-rural divide [23]. These clear-cut structural foundations of electoral alignment have led many scholars to infer electoral motives of voters in Western democratic societies directly from their material life conditions [8, 9], and to dub PRRP voters as "losers of modernization" [24], "low/medium educated" [25], "structurally threatened" [26], or in the grips of "declinism" [27].

According to some within this strand of research, anti-immigrant sentiments promoted by PRRPs are framed to appeal to those who have lost out due to globalization, usually blue-collar male workers whose jobs have been put at risk by the influx of manual immigrant workers [28, 29]. In the US, low-skilled workers were found to support restrictions on immigration more than their high-skill counterparts [30]. In the European context, studies show that individuals employed in shrinking sectors are more likely to oppose immigration than those employed in growing sectors [31], and relatedly, occupations with few exit options and low skill transfer-ability are more sensitive to potential competition with migrants [32]. In summary, the economic grievances approach highlights how social marginalization in a globalized world can amplify insecurity and vulnerability in the labor market. Globalization and modernization increase the risk of job loss, lower-paying employment, or replacement by immigrants, particularly for manual workers and those with lower education levels. Consequently, economic grievances and insecurity often fuel anti-immigrant nativism, contributing to support for PRRPs.

One could hardly deny the plausibility of explanations anchored in economic grievances, yet they are flawed in at least three ways. First, scholars have not been able to establish empirically consistent connections between individuals' economic circumstances such as income, wealth, or employment status and the propensity to support right-wing populism [33, 34]. Our results corroborate this lack of consensus. Second, criteria anchored exclusively in economic indices may not capture the social identities that individuals themselves would deem relevant, forasmuch as, according to political psychologists, individuals strive for a positive self-concept and thus tend to construct their identities in more affirmatory terms [35, 36]. Thus, support for populist causes cannot be fully, nor indeed directly, accounted for by socio-demographic marginality. Third, such ascriptive categories are also implausible since PRRPs, as a party family, do not present a coherent economic ideology or a unified stance regarding the first dimension of politics. Indeed, historically many of these parties have shifted their programmatic appeal from the 'winning formula' toward the center on economic issues [37, 38] and other parties present a flexible stance on economic issues [39]. Rather, these parties' agenda is oriented towards identity-related issues: ethnic (migration), regional (European integration), or national (minority nationalism) [40].

The cultural approach, on the other hand, attributes the rise of right-wing populism in the past decades to such trends as mass immigration and decline of traditional values [10, 34]. These developments, goes the argument, have produced a backlash, or counterreaction, particularly among white males, propelling them towards right-wing populist ideologies. However, if a counterreaction of this kind were indeed responsible for the rise of populism, we would expect the demand thereof to grow over time; yet, statistically, no such increase has been observed [41].

A growing number of scholars have used insights from both the cultural and the economic approaches to construct more complex, but potentially more causally convincing, accounts of

populism [26, 42]. Some of these explanations link changing economic conditions to right-wing populism via status anxiety: A decline in the social standing and the growing marginalization of manual workers have created a fertile ground for a politics of collective status-threat that mobilized voters leveraging their deep resentments toward professional elites and minorities [11, 43]. In this context, Guiso et al. [44] document a link between economic insecurity and distrust in political parties, on the one hand, and voting for populist parties, on the other. Erisen and Vasilopoulou (2022) found that citizens' emotional responses to perceived immigration-related threats—Specifically anger—are correlated with the support for far-right parties [45]. Rico et al. (2017) [46] show how individual levels of populist attitudes are related to emotional reactions to the economic crisis in Spain and that populist attitudes are influenced by feelings of anger rather than anxiety. Steenvoorden and Harteveld [47] demonstrate a link between support for PRRPs and societal pessimism, which cannot be explained based on objective conditions alone and finally Gest et al. [48] show that support for the radical right in the UK and the US is driven by nostalgic deprivation, the subjective perception of status decline. The above studies establish a more complexed link between economic attributes and populism, one that is mediated by negative emotions, identities or a declining trust in political institutions.

In investigating the impact of individuals' national stories on their vote choice, our paper builds on the above empirical literature that endeavors to marry the economic and cultural explanations for populism. It is, we claim, the interaction between socio-demographic attributes of marginalization, on the one hand, and national stories that pivot on the demarcation of boundaries between the in-group and out-groups, on the other, that can explain the decision to vote for PRRPs. That is not to say that the "us versus them" *Boundary* story is exclusive to populist radical right voters; to be sure, it is embraced by many others throughout the electorate. Rather, we argue that, for marginalized groups in society, this type of story is conducive to favorable self-definition, self-esteem, and a sense of positive we-ness, and that therefore, to a large extent, it shapes their voting decisions. Stories that draw a sharp distinction between one's own group and out-groups have a strong effect on people whom societal changes have left overwhelmed, disoriented, weak, or vulnerable—in other words, on those whose socio-demographic position has suffered as a result. These populations may find solace in a *Boundary* type of story because the sharp distinctions it professes allow them to attribute responsibility for their own feelings of uneasiness to factors that lie beyond their control and obligations, and thereby to maintain their self-esteem. For all these reasons, the vote of marginalized groups is affected by the "us-versus-them" *Boundary* type of story, to a greater degree than the vote of their more privileged counterparts, whose electoral choices may be guided by other considerations.

## 3. National stories and politics

The narrative mode of thought and expression is fundamental to human life. Narratives are "primary means by which individuals organize, process, and convey information" [49], and some of them are particularly instrumental in interpreting and comprehending political realities [50]. The centrality of narratives for people and groups can be epitomized as "narrative identity" [51, 52], conceptualized as "the accumulating knowledge that emerges from reasoning about our narrative memories. . . and yields a life story schema that provides causal, temporal, and thematic coherence to an overall sense of identity" [53].

Narrative research has made inroads into various fields and disciplines, including the social sciences, stimulating what is often termed "the narrative turn" [54, 55]. In the political domain this interest is evidenced in recent research spanning a range of subfields, such as policy studies

[56, 57], national security [58, 59], voting behavior [60], coalition formation [61], environment policy [62], rhetoric [63], and conflicts and their resolutions [64].

While stories that voters may carry in their minds' eye are multiple and varied, this research focuses on *national stories*–stories that are about a nation and are also embraced or shared by its people. It is the idea that stories can help us make sense of our reality and understand ourselves as political beings [50] that forms the premise of our assumption that the vote for PRRPs can be gauged through voters' stories. We focus on national stories for two main reasons. First, populist and nativist ideas in general are anchored in national identity. PRRPs, as well as their leaders, draw their support from diverse segments of the population who sometimes have conflicting interests and ideologies [65]. Rather than present coherent issue positions that may alienate some groups of voters, populist leaders use national narratives that people with diverse preferences can relate to and identify with. Second, national stories allow us to understand the ways one pictures one's nation in the *past* [66] and in the *future*, perceptions that, we claim, are liable to structure one's electoral choices in the *present*.

A number of recent empirical works demonstrate a close relationship between national stories and political behavior. In her book, Hur [67] shows that national stories that portray the relationship between the people and the nation as one of mutual commitment motivate masses of citizens to fulfill their civic duty to vote, pay taxes, or take up arms in defense of their country. In a similar vein, Shenhav and colleagues [68] found that individuals who do not hold a national story are less likely to vote than those who do. This result stands to reason: Embracing a group's social story prompts individuals to take part in that group's political activities. Shea-fer et al. [60] show that vote choices can also be predicted based on the similarity between voters' and parties' narratives. In light of the above, it is plausible to assume that individuals who are more likely to vote for populist causes might be identified based on the national stories they espouse. Accordingly, using representative surveys that measure individuals' national stories, our paper tests how the vote for PRRPs can be predicted based on national stories, even when controlling for common explanations in terms of demography and attitudes.

The concept of "story" has been defined in a variety of ways. An accepted narratological approach requires a story to contain at least two chronologically related events, either real or fictive [69]. To be sure, most stories involve more than two events, but this basic requirement allows for temporal continuity, and thus a "sequence" or "succession" [70]. In keeping with this perspective, the operational definition of "story" used here is as the chronological sequence of events derived from a narrative.

Our first theoretical expectation emanates from previous works on the importance of narratives, and specifically, on the idea that national stories reflect individuals' political identities. If so, we expect to find differences between stories embraced by voters for mainstream and populist parties. Moreover, the demand side literature describes voters for PRRPs as having reverted to nativist values of exclusion and discrimination, in opposition to elites' global and universalistic values [71], and as tending to glorify the past [66]. While populism and exclusionary nationalism, embodied in the Boundary type of national story, are analytically distinct constructs, they both play roles in the populist radical right. Populism revolves around the 'people as underdog' dynamic along an up-down axis, while exclusionary nationalism focuses on the 'people as nation' along an in-out axis. Despite their differences, both are integral components of the populist radical right. Previous works claim that right-wing populist parties mobilize potential voters by fueling feelings of resentment and anger, often leveraging "us-versus-them" dichotomies, whether between "the people" and the "corrupt elite" or through negative sentiments toward out-groups [72–74].

A key feature of the PRRP ideology is nativism, a belief that "states should be inhabited exclusively by members of the native group ("the nation") and that nonnative (or "alien")

elements, whether persons or ideas, are fundamentally threatening to the "homogeneous nation state" [75]. Indeed, the distinction between "us" and "them" is at the ideological core of PRRPs and is touted by populist leaders, who shift all blame from "the innocent people" to either out-groups or "corrupt elites."

Notably, however, while advocating an "us vs. them" story, populist parties focus mainly on current politics. Stories, on the other hand, are least of all about current politics: They weave together the nation's past events with future events regarding it, and as such effectively constitute a blueprint for how people perceive their nation. Rooted in a very broad perception of the nation, a national story reflects the national identity of the group that embraces it. Accordingly, a national story does not deal with current policy and occurrences, which preoccupy political parties.

Hence, we hypothesize that:

*H1*: National stories of PRRP voters will differ from those of mainstream party supporters. Specifically, holding a *Boundary* story that professes a clear distinction between "us" and "them" increases the likelihood of voting for a right-wing populist party.

## 4. Why are some voters more sensitive than others to a *Boundary* story?

The above assumptions do not imply that all voters who hold an "us versus them" *Boundary* story are equally likely to vote for a populist party or leader. Individuals differ as to the facility with which they translate this kind of national story into voting for PRRPs. What explains this difference in the effect of a *Boundary* story on vote choice? Our answer takes count of economic grievances: We make a case for an interaction effect between one's national story and one's structural socio-demographic positioning.

We argue that a *Boundary* story is likely to be more central to the self-definition of individuals stationed along an in-group's periphery, in terms of geography (far flung), education (medium and low) and status (low), and are in a sense marginalized. Some studies define marginalization in objective terms, as belongingness to a group that is deprived of certain resources [48, 76–78], while others define it as the subjective sense of low societal status or lack of recognition [11, 79–81]. Yet, while these studies delve into different facets of "left behindness," they do not directly pinpoint the primary motivator behind support for PRRPs. We propose that socially marginalized individuals, who recognize themselves as belonging to low-status groups, are often driven to exclude others to enhance their own sense of belonging within a community [82]. In other words, marginalized people might cope with their subordinate position and at the same time strive to gain greater acceptance to mainstream society by distancing themselves, physically or psychologically, from other marginalized groups, and labeling the latter as outsiders and a threat to the community. These "left-behinders" might be similar or proximate to some of the out-groups they denigrate, such as immigrants, in terms of marketable skills, status or education. Yet, they take great pains to distinguish themselves from them in an effort to affirm their in-group's uniqueness [83] and to bolster their own positive sense of collective worth, which buffers against their stigmatization [84].

Accordingly, adherence to a *Boundary* story should not be as important to the self-definition and self-esteem of more privileged voters. We argue, therefore, that to explain populist voting, it is necessary to take count of an interaction effect between the "us versus them" *Boundary* story and socio-demographic attributes such as education, social status and geographic periphery. Those who have lost out to globalization and whose socio-economic

resources are limited will be more willing to translate their *Boundary* story into vote choice than their counterparts with better resources and socio-demographic positioning. Hence, our second hypothesis:

*H2*: The positive correlation between the *Boundary* story and support for PRRPs will be stronger within marginalized communities compared to their more privileged counterparts.

This hypothesis aligns with the economic grievances literature, arguing that socio-demographic attributes are important for populist support, but crucially adds an identity or cultural aspect—the type of national story one holds. We hypothesize that it is the combination of holding a *Boundary* story with belonging to a socially marginalized group that is positively associated with the vote for PRRPs. In the next section we empirically examine these hypotheses.

## 5. Empirical strategy

### 5.1 Data and measurement

The study tests the relationship between right wing populist voting and national stories in four countries with different histories—notably, with respect to immigration and PRRP support: Denmark, the Netherlands, the United Kingdom and the United States. These countries were chosen also because they are all developed democracies, but differ in their political systems as presidential vs. parliamentary democracies, as well as their electoral systems and district magnitude (e.g., extreme proportionality in the Netherlands vs. single-member plurality in the UK and the US). The above variations across the countries sampled make it possible to extrapolate the results to other developed democracies and draw general inferences regarding the relationship between national stories and vote choice.

In the above four countries we administered large-scale representative online surveys among citizens with different background characteristics (e.g., low education level or political alienation). Respondents in all countries were asked the same core set of questions, including two items tapping past and future components of their respective national stories as proxies for the entire story. In what follows, we analyze and compare the story profiles of voters for populist and for other, mainstream parties.

The four surveys were administered online in the respondents' native language. The Danish survey ($N = 1,010$) was conducted in June 2015, during that year's parliamentary election; the Dutch survey ($N = 1,448$)–during the 2012 parliamentary election; the British survey ($N = 1,002$)–during the 2015 parliamentary election; and the American survey ($N = 1,001$)–during the 2016 presidential election (see S1A Appendix).

**5.1.1 Dependent variable: Voting intention.**   We measured respondents' voting intention by asking them the following question: "If parliamentary (presidential in the US) elections were held today, for which political party (candidate in the US) would you vote?" In each country, this question was followed by a full list of parties (or candidates) running in the respective election. To avoid order effects, the sequence in which the names of the parties/candidates were presented was randomized.

Based on respondents' answers to this question, we constructed a dichotomous variable in which populist vote was coded as 1 and any other—as 0. Relying on the ideational approach to populism that understands politics as a Manichean struggle between the will of the homogenous people and the corrupt elite and which argues that political sovereignty should reside with the ordinary people [4, 18], we utilize the PopuList data to classify parties as populist and radical right parties. The PopuList is based on country-specific experts' and comparativists' assessment of parties' core ideological attributes. For the US case we rely on Inglehart and

Norris's classification [34]. In the US, this dichotomous variable differentiated between the intention to vote for Donald Trump (populist vote) and any other candidate. To assess the robustness of our results, we also constructed a three-pronged variable differentiating between populist, mainstream right, and mainstream left voting intentions. Our classification to the mainstream right and mainstream left party-families was based on two comparative datasets: Laver, Gallagher, and Mair [85], and Parlgov [86]. In Denmark, mainstream right parties were the Conservative People's Party and the Liberal Party (Venstre); mainstream left parties were the Social Democrats, the Socialist People's Party, Unity List, and the Social Liberal Party; and the populist party was the Danish People's Party. In the Netherlands, mainstream right parties were Christian Democratic Appeal and the People's Party for Freedom and Democracy (VVD); mainstream left parties were the Labour Party (PvdA) and the Socialist Party (SP); and the populist party was the Party for Freedom (PVV). In the UK, the mainstream right party was the Conservative Party; mainstream left parties were the Green Party and the Labour; and the populist party was the UK Independence Party (UKIP). We report the results of these alternative models in S1E Appendix.

**5.1.2 Independent variable: National story.** Our proxy for national stories is based on open-ended questions where responses are not restricted to a predefined set of categories [61, 68]. Specifically, to tap respondents' most important *past event*, they were asked: "When you think about the history of [country] and the [American, British, Danish, or Dutch] people, which *past event* do you consider to be the most important?" To measure perceptions of the most important *future event*, we asked: "When you think about the prospects of [country] and the [American, British, Danish, or Dutch] people, what *future event* do you wish to see?" These two open-ended questions directly target the nation's past and future, and thus tap by proxy respondents' national stories. After recording respondents' answers to the two questions above, we grouped the answers into categories that contain references to substantively identical or highly similar events. For example, identical answers such as "WWII" and "Second World War" were grouped into a single "World War II" category, and highly similar answers such as "When Winston Churchill was Prime Minister" and "Churchill's leadership" were combined into a category labelled "Churchill's era." To ensure that our categorization of respondents' answers was reliable, two independent coders were asked to assign 300 past and future answers to categories based on a list of all coding categories. Intercoder reliability was Krippendorff's alpha = .83 or higher for the past category, and .7 or higher for the future category.

In this research two sets of categorizations were applied to the national story variable. The first was adapted from the Comparative Agenda Project (CAP): Past and future components of the national stories were relegated to 18 policy categories. The objective was to reduce variation when testing the association between stories and vote choices (see Figs 1–4). Next, we sorted all past and future events to three story types, termed here as *Survival*, *Self-expression*, and "us versus them" *Boundary*. These three categories mirror broad political divisions in advanced democracies. The first two are based on the work of Inglehart and Welzel [87, 88] and of Norris and Inglehart [7] in the World Value Survey, in which people's value priorities are aggregated at the country level, and according to these aggregations, each country is then placed on a two-dimensional value continuum. On the first axis, self-expression values are at one pole and survival values at the other. Self-expression values emphasize autonomy, human and minority rights, freedom, environmental protection, and quality of life, whereas survival values center around economic and physical security [88–90]. The *Boundary* and *Self-expression* story types also tap important political divisions in affluent societies, usually termed as the universalism-particularism or cosmopolitanism-communitarianism divide [91]. *Boundary* national stories establish a division between "the people," or the national community–"us"– and everyone outside it–"them." These stories inherently position other nations as outgroups,

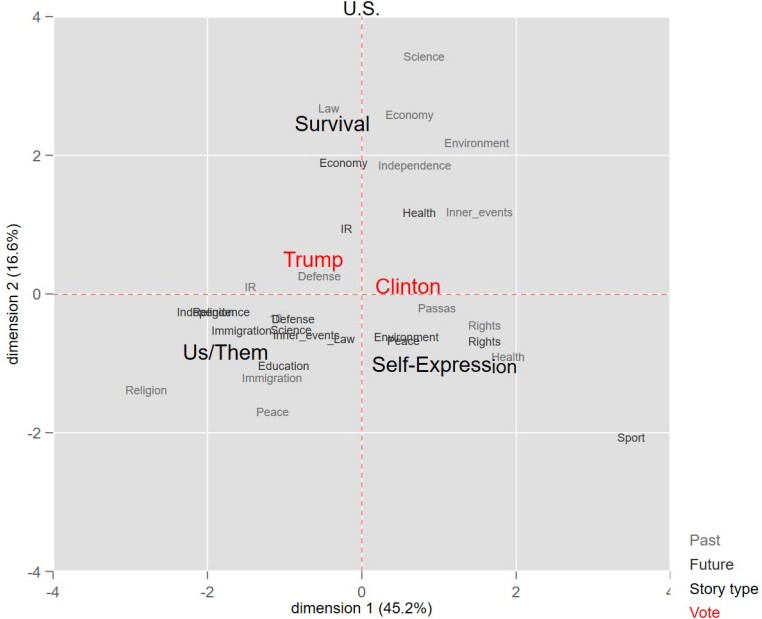

**Fig 1. Multidimensional correspondence analysis of national stories and vote in the US.**

emphasizing a separation between them and the ingroup (the nation). The "us-them" boundary is thus a site for conflict between the national community and the outgroups. According to our coding scheme, responses relating to conflicts and clashes between the nation and an outgroup were coded as *Boundary*.

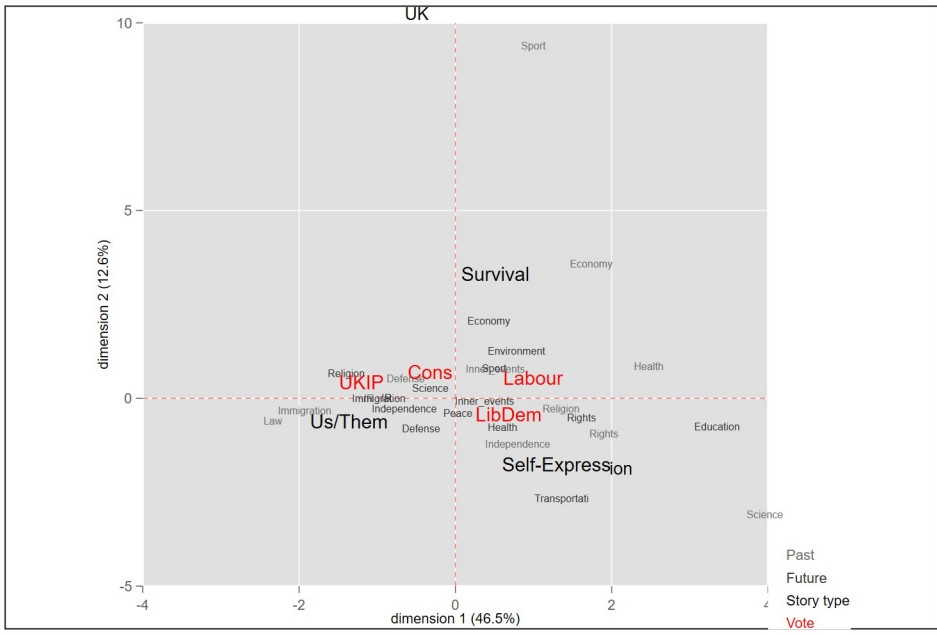

**Fig 2. Multidimensional correspondence analysis of national stories and vote in the United Kingdom.**

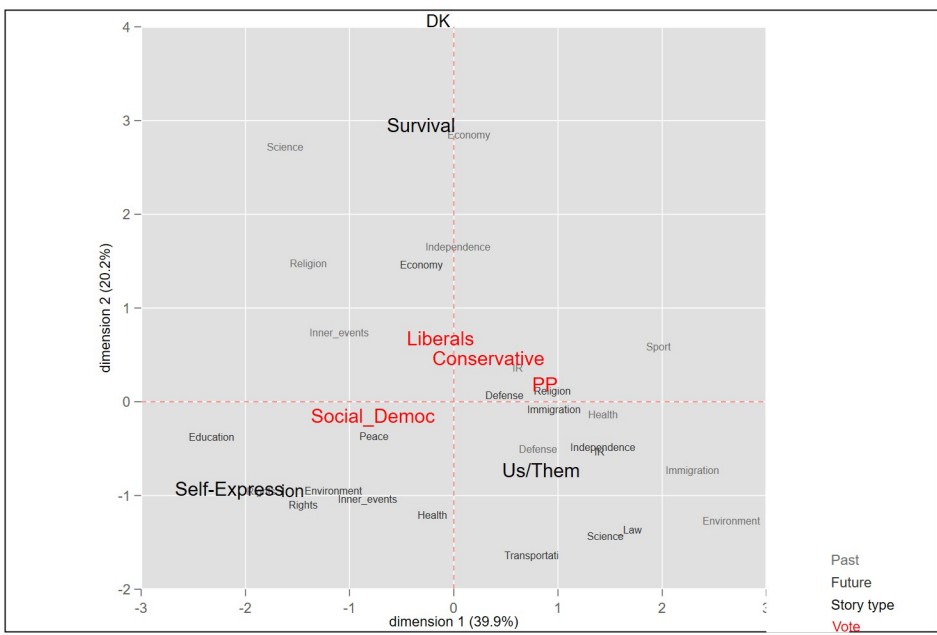

**Fig 3. Multidimensional correspondence analysis of national stories and vote in Denmark.**

*Survival* stories, on the other hand, center around the nation's economic and physical security and incorporate events and issues such as "the Industrial Revolution" and "Napoleonic wars" for the past, and "job opportunities" and "dealing with terrorism" for the future. Some events or issues lend themselves to a classification as both *Survival* and *Boundary*, and were

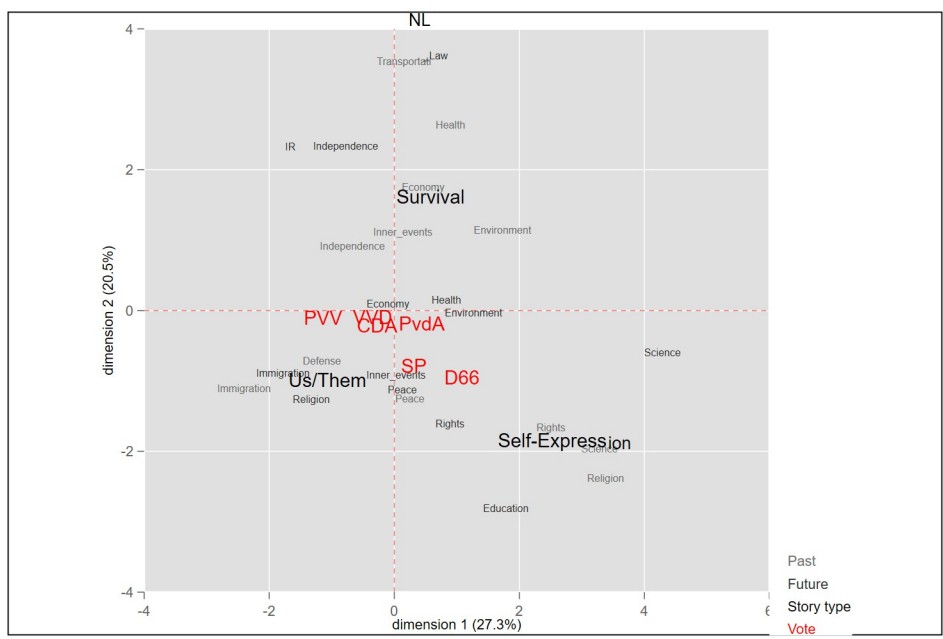

**Fig 4. Multidimensional correspondence analysis of national stories and vote in the Netherlands.**

accordingly relegated to both rubrics. For instance, narratives about wars or conflicts, such as "World War II," are salient to the nation's physical security while at the same time marking its separation from other nations. Such elements were entered under both the *Survival* and the *Boundary* rubrics. *Self-expression* stories revolve around human rights, democracy and the quality of life with stories such as "women's rights," "peace," "signing the Magna Carta," "suffrage," and "freedom" are coded as such.

The three categories, Survival, Self-expression, and Boundary are not mutually exclusive; a story event can be classified under multiple types. For instance, World War II, a common past event, is classified as both survival and boundary. Importantly though, not all survival stories are boundary, and vice versa. This distinction is evident in our regression results, where survival tends to predict voting for mainstream parties, while boundary predicts voting for PRRPs. Our classification is just one step in constructing the overall story. The second step involves combining past and future story components into a cohesive narrative that each individual holds. Respondents receive scores on the three indices of story types based on their composite narratives. For example, a respondent with a World War II past event and a peace future event would score 1 on all three indices, indicating elements of survival, boundary, and self-expression. Another respondent with a World War II past event and an economic prosperity future event would score 2 on survival, 1 on boundary, and 0 on self-expression.

It is important to note that our classification pertains to the stories themselves, not the individuals holding them. We use a 'text-oriented' approach to reading the text rather than 'author-oriented' since we cannot determine the individual intentions of every respondent or the feelings these stories evoke. While such reading of the text has its pros and cons, it makes sense to assume that if a person orients her national stories around survival-related content (e.g. economic or conflict-related elements), even historical ones, it suggests a priority or at least an attention to survival framing, in comparison with respondents who chose events relating to, for example, women's rights. A detailed explanation of the coding implemented, along with intercoder reliability scores and descriptive statistics for the three story types, are provided in S1B Appendix. Table 1 displays the most common past and future story components for each country by story type. S1C Appendix displays the five most common past and future concerns for each country.

The score on the boundary, survival and self-expression scales was assigned based on the number of components a respondent provided for the respective story type. Thus, for example two survival components (one for the question about the past and one for the question about future prospect) earned the respondent the highest score on that scale (2), one component (past or future) earned the respondent a mid-score (1); while absence of such components was tantamount to the lowest score (0). The higher the score on a scale, the stronger the leaning towards the respective national story type. A similar score was produced for the other types of national stories (Correlation between the three story-type scales is low to medium, and spans from .42 between the boundary and survival scales in Denmark to -.09 between the survival and self-expression scales in the UK).

Note that our categorization scheme does not encompass all story components cited by respondents, such that some were left uncategorized. Examples of story components that are assigned a zero score on all scales are as follows: In Denmark, "The Christianization of Denmark" or "Winning the EU football championship," both for the past; in the Netherlands, the "death of Prince Claus" for the past and "political stability" for the future; in the US, "the Hillary Clinton era," "Kennedy's assassination," and "slavery"–all for the past. The proportion of this left-out group is 26% in Denmark, 22% in the Netherlands, 28% in the US, and 21% in the UK.

*5.1.2.1 Control variables.* The attitudinal and demographic variables we controlled for were derived from prior research on populist voting. The attitudinal covariates are political

**Table 1. The most common past (p) and future (f) story components by story type.**

| Country/Story | Boundary | Self-expression | Survival |
|---|---|---|---|
| DK | F: Immigration policy<br>Out of EU<br>Dealing with terror<br>P: WW2<br>Getting\losing Southern Jutland | F: Integration and tolerance<br>Peace<br>Good education<br>P: Women's rights<br>The introduction of democracy | F: Employment opportunities<br>Strong economy<br>Tax reform<br>P: WW2<br>Getting\losing Southern Jutland |
| US | F: End of terror<br>P: September 11 attacks<br>Civil war | F: Peace<br>Better education<br>P: Freedom, rights, equality | F: Better economy<br>More jobs<br>End of terror P: September 11 attacks<br>Civil war \| |
| UK | F: leaving the EU<br>Immigration policy<br>Reducing the power of the EU<br>P: WW2<br>Napoleonic wars<br>Norman Conquest 1066 | F: Equality<br>Peace<br>A fairer society<br>P: Signing the Magna Carta<br>Women's rights Parliamentary democracy | F: Strong economy<br>job opportunities<br>End poverty<br>P: WW2<br>Norman Conquest 1066<br>Napoleonic wars |
| NL | F: Leaving the EU<br>limiting immigration<br>anti-Islam<br>P: WW2<br>War in Afghanistan | F: Better tolerant society<br>freedom<br>civil rights<br>P: Suffrage<br>Emancipation<br>Secularization<br>Enlightenment | F: Economic growth<br>solve economic crisis<br>P: War in Afghanistan<br>WW2 |

Having sorted past and future story components to story types, we constructed three scales, one for each type.

knowledge, political interest, and ideological self-placement. The demographic variables are gender, age, education, social class, and a dummy variable for living in a rural region. In S1E Appendix we also control for populist attitude. The operationalization of all control variables, as well as their descriptive statistics, per country, can be found in S1D Appendix.

**5.1.3 Models and estimation.** To assess the differences in national stories of voters for populist versus mainstream parties (H1), we first ran a Multidimensional Correspondence Analysis (MCA) for each country separately. This analysis spatially locates the national stories and the vote choice variables on a two-dimensional coordinate graph, thus representing the association visually. The smaller the Euclidian distance between answer categories of the different variables, the more correlated they are. Next, to strengthen these story-vote associations found in the raw data, we ran logistic regressions to predict, for each respondent, their intention to vote for a populist party as a function of their scores on the three story-type scales. In the third analysis, we examine whether the effect of stories on the vote is moderated by respondents' socio-demographic attributes (H2). For that purpose, we predict the vote for PRRPs as a function of national stories, demographic attributes, and the interaction of the two.

# 6. Results

## 6.1 Different stories, different vote choices

Our first hypothesis states that populist voters' story components will have some distinctive characteristics and that their national story will differ from those of voters for mainstream parties. Empirically, we expect populist voters' stories to be spatially separated from the ones of mainstream voters. We gauged the differences in the story-profiles of these two populations by means of Multidimensional Correspondence Analysis (MCA). MCA is a parsimonious method for mapping relations that are active within a dataset by representing the distances

between categorical variables in a smaller space. These relations are summed up as a two-factorial space that can map the greatest amount of the information contained initially in the dataset, thus rendering MCA a well-suited method for evaluating story proximity [61]. The analysis searches for a space of two dimensions for representing the maximum association between response categories of the variables included in the analysis. The smaller the Euclidian distance in the MCA plots between response categories, the greater the similarity between them, and vice versa.

We apply MCA for each country separately and identify the relationships among (1) respondents' voting intentions, (2) their reported most significant national past event, and (3) their reported future events they wish to see for their country. By incorporating these three variables into the analysis, we can pinpoint associations between past and future components of the national story and vote choice. Thus, MCA enables us to examine H1, i.e., whether and how national stories differ between voters for mainstream and populist parties.

Figs 1–4 is a graphic display of the MCA results. It shows, first, that in all countries, the three story-types–"us vs. them" *Boundary*, *Survival* and *Self-expression*–are located far from one another, each in a different quadrant. Clearly, these story types represent different perceptions of the nation: what it used to be and where it should head in the future. Second, Figs 1–4 show a close proximity between the vote for a right wing populist party and the *Boundary* story across all four countries, indicating a correlation between holding this type of story and voting for a populist party. Third, the *Boundary* and the *Self-expression* stories are located at the opposite poles of the first, horizontal, axis (thereby accounting for most of the variance), and thus stand in competition. On the other hand, the *Survival* story, which centers mainly on economic and security issues, is located on the vertical axis, and explains less variance. This could suggest that the *Survival* story is less pertinent to partisan divisions, compared with the other two.

Overall, the past and future story components cited by voters for PRRPs tend to differ from those of mainstream voters, such that the categories of *Immigration*, *Law*, *Religion*, *Independence* and the like are located closer to the populist vote, while *Peace*, *Health*, *Rights*, *Sport*, *Education* and *Science*–further from populist and closer to mainstream parties by a large margin. In the US, the horizontal axis captures more than 45 percent of the associations in the data: voters for the Democratic Party (Clinton) are located right, while Trump's voters—left to the origin. Looking away from Figs 1–4, at the actual stories extracted from the raw data, the most prominent past events cited by Trump voters are the *September 11 attacks*, the *Constitution*, and *Independence*, while for Clinton voters, these issues are *Freedom*, *Rights and Equality*, and the *Civil War*. Trump voters' events they wish to see for the future relate to *Immigration*, *Better economy*, and *Election results*, while Clinton's supporters are concerned with issues of *Freedom*, *Rights and equality*, *Better economy*, and *Peace*. The picture that emerges in the US case is that of a polarized society, in which national story profiles are categorically different for the two political sides.

In the UK, the situation is somewhat different. UKIP and Conservative voters are not far apart. Both these groups appear in the second quadrant, with the UKIP closer to *Immigration* and *IR*, while Conservatives, to *Independence* and *Security*. The story profile of UKIP voters is different from those of others: the most dominant past concerns they cited pertain to *WWII*, *Immigration*, and *Norman Conquest*, while for mainstream voters (Labour and Conservative) these were *World War II*, *Women's rights*, and *The establishment of the National Health System*. As regards future events for their country, voters for UKIP and for mainstream parties converged in citing *Immigration* and *Strong economy* the most frequently but diverged with regard to *Exit the EU* and *Equality*, the former predominant among the UKIP while the latter among mainstream voters.

In Denmark, the past events cited the most frequently by voters for the Danish People's Party fall under the rubrics of *Immigration*, *Law*, and *Independence*. The two most dominant past events of DPP and mainstream voters are the *Introduction of the Basic Law* and *WWII*. Another past issue central in the mind's eye of DPP voters is *Losing Southern Jutland*, while mainstream voters consider *Women's rights* of central importance. Voters for populist and mainstream parties share one dominant future aspiration, *Improved welfare system*, but differ on others, with mainstream voters dwelling on *Environmental policy* and *Employment opportunities*, while populist voters on *Immigration* and *Exiting the EU*. In the Netherlands, the events most frequently cited by the Party for Freedom (PVV) voters in regard to the past are *The assassination of Pim Fortuyn* and *The Eighty Years' War*, while in regard to the future most concerns revolve around *Exiting the EU* and *Restricting immigration*. For the CDA mainstream right voters, these are, respectively, *WWII* and *Independence*, and *Better tolerant society*, *Further deepening European integration*, and *Economic growth*. For mainstream left voters (PvdA) these are, respectively, *WWII* and *Suffrage*, and *Better tolerant society*, *Economic growth*, and *Equal distribution*.

The analyses presented above indicate a link between national stories and vote choices across all four of our cases. In what follows, we estimate separately for each country a vote choice regression model as a function of the three story-type scales.

## 6.2 National stories as predictors of the vote

We have assumed that support for populist parties could be predicted by the three scales we constructed, each capturing a different national story type. Specifically, according to H1, individuals who embrace an "us-versus-them" *Boundary* story will be more likely to vote for right wing populist parties. Furthermore, H2 posits that this association is moderated by socio-demographic characteristics, such that the effect of this story type is stronger among marginalized groups.

To further test H1, we gauged the effect of the three story types on the vote for populist parties by estimating logistic vote choice models, which are presented as coefficient plots in Fig 5. In the models, we compare respondents' propensity to vote for a right wing populist party as opposed to any other party in the country. The coefficients (including CIs) of our main independent variables are presented in Fig 5. Importantly, in the regression models we control for political knowledge, political interest, and ideological self-placement. in S1F Appendix we reran the regressions controlling for populist attitudes as a robustness check. The models also incorporate a set of demographic controls: gender, age, education, social class, and rural/urban divide.

Across all four cases, the "us versus them" *Boundary* story has a positive and significant effect on the vote for populist parties/leaders, while the *Survival* and *Self-expression* stories have a negative, and in some cases significant effect on the populist vote. The magnitude of the effect is substantial. In Denmark, the UK, the US and the Netherlands, moving from not citing either past or future *Boundary* story components to citing both such components increases the probability to vote for the right-wing populist party/leader by 20, 19, 20 and 15 percentage points, respectively. No consistent effect for the socio-demographic variables (social class, education, and rural/urban divide) was revealed—in line with the empirical literature on the economic-grievances explanations. In S1E Appendix, we offer a somewhat different modeling strategy: instead of a logistic vote-choice model predicting the vote for populist parties, we estimate multinomial vote-choice models where the dependent variable is three-pronged (mainstream right, mainstream left, and a right wing populist party, with intention to vote for mainstream left as the reference category). Our results hold.

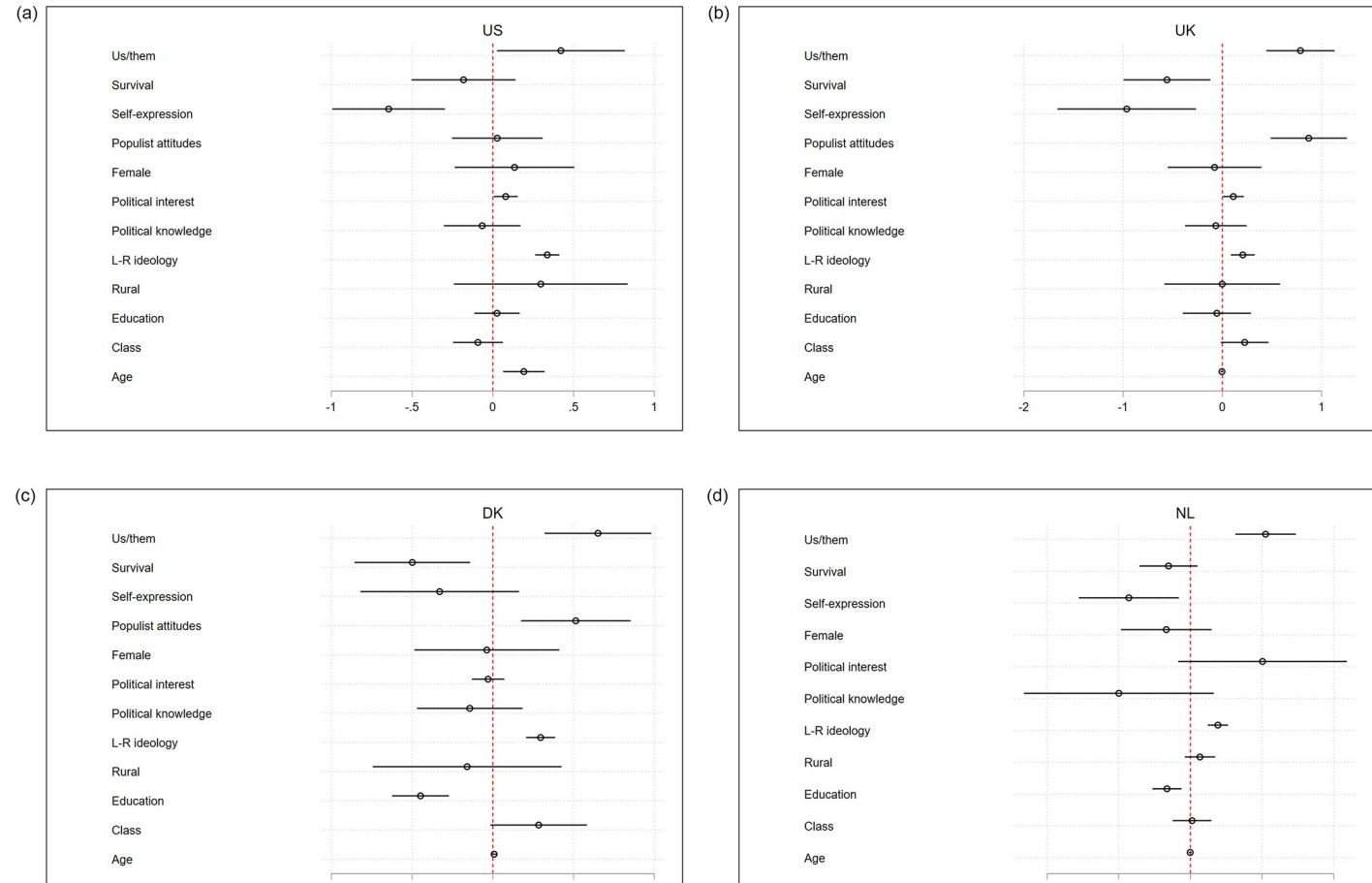

**Fig 5. The effect (displayed as coefficients) of respondents' national stories on their vote choices.** *Note*. The *boundary*, *survival* and *self-expression* are three scales for the three national story types. Regression models include socio-demographic controls and attitudinal variables (political interest, political knowledge and ideological self-placement).

## 6.3 A similar story yet different vote

Taking stock of our findings thus far, it seems that not only are national stories predictive of electoral behavior (even when controlling for variables that are known to explain voting for PRRPs), but it is a specific type of story that correlates positively with the vote for right wing populist causes—one pivoting on in-group favoritism and out-group hatred, the "us versus them" stance, and on demarcating boundaries between the community and other groups, inside and outside the nation. But is the *Boundary* story exclusive to the electorate of populist radical right parties? Is it possible that some of the voters who support mainstream parties also adhere to that type of national story? And if so, what diverts them from supporting PRRPs? In other words, why does embracing an "us versus them" *Boundary* national story not entail voting for populist parties across the board?

Our second hypothesis points at a differentiated effect of the national story on the vote. We hypothesized that marginalized groups are more prone to translate the exclusionary *Boundary* story into voting decisions, while its effect on the vote of their better-off fellow citizens is less potent.

Fig 6 presents the share of voters who hold an "us versus them" *Boundary* story among populist, mainstream right and mainstream left parties. While a larger share of the populist electorate embraces such a story, it is also the case that among the mainstream right, and to a lesser extent, the mainstream left, there are voters holding this story as well. Who are these mainstream right and left voters and why aren't they acting upon their vision of the nation? Why is their vote shielded from the electoral effect of the exclusionary story? To answer these questions, for each country separately, we interacted the "us versus them" *Boundary* story variable with different socio-demographic variables (education, social class, and urban/rural divide).

Results are shown in Fig 7, which displays the marginal effect of the "us versus them" *Boundary* story on the vote for right wing populist parties [92] across different levels of education, social class, and residence (rural/urban). Regression tables are presented in S1F Appendix. The downward trend of all graphs implies that the "us versus them" *Boundary* story has a positive effect on the vote for PRRPs among the so-called "left-behinders," or marginalized groups —the low educated, the poor, and residents of rural/peripheral areas. However, this effect diminishes with the progression to the right on the horizontal axis, such that no effect of an "us versus them" *Boundary* story is found among privileged voters: the highly educated, the wealthy, and those who live in big cities. As for the urban/rural electoral divide, indeed it was found to be is highly pronounced in the majoritarian democracies of the UK and North America, where the republican and conservative parties rely on support from rural areas, but it is also evident in Denmark and to a lesser extent in the Netherlands [93].

Due to space limitation, only substantive and significant results are displayed in Fig 7. The full results of this empirical exercise are presented in Fig F1 in the S1 Appendix. Fig F1 in S1 Appendix presents the interaction effects between all the socio-demographic variables and a Boundary story. No statistically significant results were obtained for the urban/rural divide in the UK, for social class in the Netherlands, and for education in the US and Denmark.

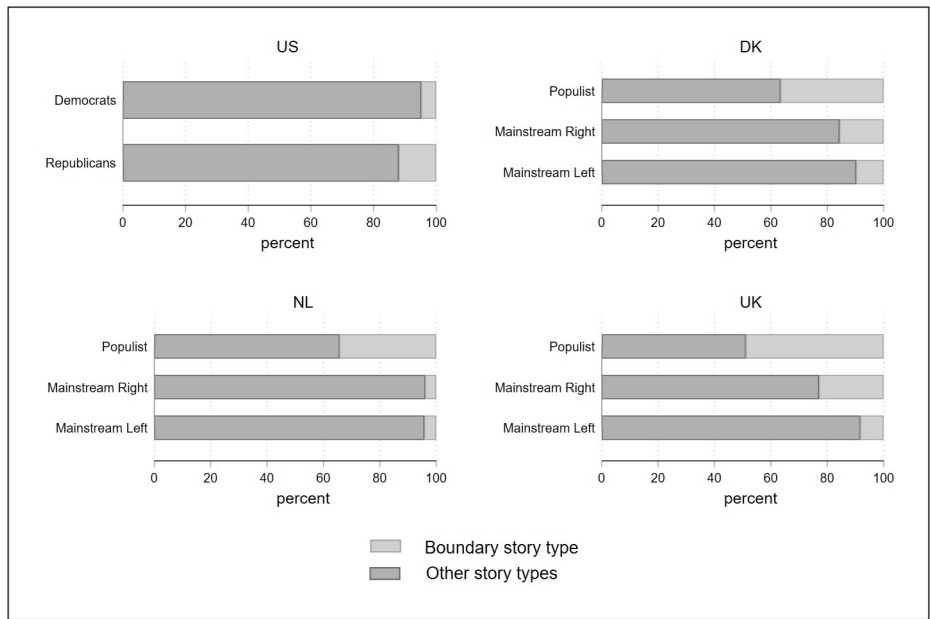

**Fig 6. The "us versus them" Boundary story among voters for mainstream and populist radical right parties.** *Note.* Fig 6 displays the dispersion of the "us versus them" *Boundary* story type among populist and mainstream voters in each of the four countries studied.

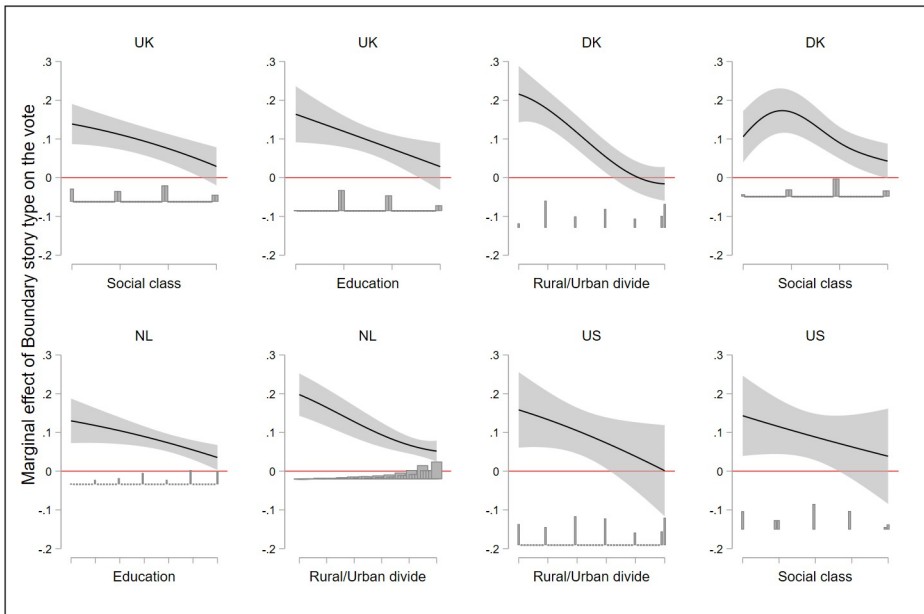

**Fig 7. Estimated effect of the "us versus them"** *Boundary* **story type on the vote for populist radical right parties.**
*Note*. Marginal effect of the "us versus them" *Boundary* story type on the vote for populist radical right parties (vertical axis) across levels of different socio-demographic variables. Marked are 95% confidence intervals. All moderating variables (on the x axis) span from marginalized groups to more privileged voters: The social class variable spans from working to upper class; education spans from 8th grade or less to graduate and post graduate; and Urban/Rural spans from living in a rural area to living in a city/town with over 1,000,001 inhabitants. Results are based on estimation reported in S1F Appendix. The histogram below the predicted margins displays the distribution of the moderators. Figure uses the INTERFLEX package [93].

## 7. Conclusion

The literature on the demand side of right-wing populism has grappled with the question regarding the relative role of economic and cultural factors, and how they might be linked. Our study feeds into this fundamental debate by exploring the link between voters' socio-demographic attributes and their national identity couched in the national stories they embrace. This is the first study to probe the relationship between voters' narrative perceptions of their nation and support of PRRPs. Not only did we find that national stories espoused by PRRP voters differ from those of mainstream party supporters, but also that the "us versus them" *Boundary* story type predicts the vote for PRRPs, and that it is salient chiefly for the vote of marginalized groups in society.

Our study opens up exciting new research avenues, and three of these are particularly note-worthy. To begin with, future research can explore the mechanisms underlying our main find-ings. Our results are based on correlational data, while face-to-face interviews or experiments can help in unraveling the reasons why marginalized individuals who hold an "us versus them" *Boundary* type of story act upon it at the ballot box, whereas their better-off counter-parts who espouse the same kind of story do not. The impact of identity rooted in an "us versus them" *Boundary* story can be causally examined by experimentally manipulating this variable in terms of the strength of conviction. Additionally, panel studies can shed light on changes in a society's national stories over time. Furthermore, research has shown that voting for populist parties does not always align with populist attitudes, and that some voters with populist views may choose mainstream parties for strategic or other reasons; accordingly, future research

could explore the connection between national stories and populist attitudes. Our findings indicate that an "us versus them" *Boundary* story is prevalent not only among populist voters but also among those supporting mainstream parties. A follow-up study could provide a more nuanced understanding of the association between the "us versus them" *Boundary* story and populism, measured in terms of attitudes rather than voting choices.

Another extension of our research involves national stories as dependent variables. In the present study, national stories figure as independent variables predicting the vote for PRRPs. It is our opinion, however, that national stories are not fixed entities. They are continuously constructed, reconstructed, and modified by political parties and leaders, who invoke them to mobilize followers. More theoretical and empirical research is needed to elucidate the causal relationship between national stories, vote choice and party identification. Are the former two factors affected by the latter, or vice versa? Or perhaps the relation is reciprocal?

A third potential extension to this study concerns our proxy for national stories. Our current method, employing past and future events to portray the national story, can be further developed. Future studies could enhance and expand upon this approach. For instance, broadening the range of events, or incorporating additional elements such as specific sentiments towards events could yield a more comprehensive and nuanced understanding of the associations between national narratives and populist attitudes. Since our focus was on analyzing respondents' perceptions of their nation and their relationship with voting behavior, this study primarily focuses on national stories. However, future research could explore the impact of populist stories on voter choice by aligning populist elements, such as people centrism and anti-elitism, with individuals' narratives.

The findings reported in this paper document the power of national stories to shape vote choice in different political contexts. By learning peoples' preferences in regard to their nation's past and future, one can determine their electoral choices at present. Such perceptions constitute a powerful theoretical and empirical tool that deserves greater attention from political behavior researchers than it has received to date.

## Supporting information

**S1 Appendix.**
(DOCX)

## Acknowledgments

We would like to thank Tamir Sheafer, Alon Zoizner, Anita van Hoof, Jan Kleinnijenhuis, Yael R. Kaplan, David Nicolas Hopmann.

## Author Contributions

**Conceptualization:** Odelia Oshri, Eran Amsalem, Shaul R. Shenhav.

**Data curation:** Shaul R. Shenhav.

**Investigation:** Odelia Oshri, Eran Amsalem, Shaul R. Shenhav.

**Methodology:** Odelia Oshri.

**Writing – original draft:** Odelia Oshri, Eran Amsalem.

**Writing – review & editing:** Odelia Oshri, Shaul R. Shenhav.

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
