## [Decision Letter · Decision Letter 0]

18 Oct 2023

PONE-D-23-23023Voices from the margins: How national narratives are linked to support for populist partiesPLOS ONE

Dear Dr. Oshri,

Thank you for submitting your manuscript to PLOS ONE. After careful consideration, we feel that it has merit but does not --at this stage-- fully meet PLOS ONE’s publication criteria. Therefore, we invite you to submit a revised version of the manuscript that addresses the points raised during the review process. Both reviewers find this paper interesting. Whilst some of the comments are major, a few of them can be considered as minor. So, please note that providing a revision does not necessarily guarantee an acceptance of publication in the next round. Here are a few important points raised by the reviewers:  Both reviewers recommend that there is a need for more emphasis on "nativism" rather than "populism," as the paper deals with right-wing parties. Similarly, both agree that the paper lacks clarity in explaining the generation of national story variables, including whether they are on a continuous scale or categorical. You should also clarify how you classify parties with additional explanation (R1). Important in this conceptual discussion is the point made by R1 regarding the “us versus them” narrative. I think you need to carefully address this concern and make sure that your theoretical approach is solid for the reader.  

Regarding the empirical assessments, you need to provide the rationale for selecting Denmark, the Netherlands, the UK, and the US as case studies (R2), as well as explain the inconsistencies across your models (R1). Robustness tests would be appreciated.

Also, note that you need to consider voting for populist parties doesn't necessarily equate to having populist attitudes, and some voters might opt for mainstream parties. This distinction should be acknowledged and relevant work in the field could be referred. In that, try to find out work that focus on populist attitudes as dependent measures in comparison to work that focus on populist party vote choice. 

We look forward to receiving your revised manuscript.

Kind regards,

Cengiz Erisen

Academic Editor

PLOS ONE

Journal Requirements:

3. Please ensure that you include a title page within your main document. We do appreciate that you have a title page document uploaded as a separate file, however, as per our author guidelines (http://journals.plos.org/plosone/s/submission-guidelines#loc-title-page) we do require this to be part of the manuscript file itself and not uploaded separately.

6. Please amend your list of authors on the manuscript to ensure that each author is linked to an affiliation. Authors’ affiliations should reflect the institution where the work was done (if authors moved subsequently, you can also list the new affiliation stating “current affiliation:….” as necessary).

Reviewers' comments:

Reviewer's Responses to Questions

**Comments to the Author**

1. Is the manuscript technically sound, and do the data support the conclusions?

Reviewer #1: Partly

Reviewer #2: Partly

2. Has the statistical analysis been performed appropriately and rigorously? 

Reviewer #1: No

Reviewer #2: Yes

3. Have the authors made all data underlying the findings in their manuscript fully available?

Reviewer #1: No

Reviewer #2: Yes

4. Is the manuscript presented in an intelligible fashion and written in standard English?

Reviewer #1: Yes

Reviewer #2: Yes

5. Review Comments to the Author

Reviewer #1: (FULL REVIEW UPLOADED AS AN ATTACHMENT)

The paper presents an interesting and potentially novel idea. I personally liked the idea of testing the effect of "narratives" (called here "national stories") on the populist vote, and I think there is much potential in this paper.

However, in the end, these are not adequately explored, and the authors overestimate their contribution. In general, some fundamental flaws make its publication not recommended. Namely, major problems with its research design, theory, and writing require substantial rework and reconsideration. Broadly speaking, these are:

1 – Regarding their research design, the authors propose three research questions but can only answer (and hypothesize on) two.

2 – In their research design and results, the authors argue that there are significant differences in the national stories embraced by voters of PRRPs compared to those of mainstream party supporters (introduction, pages 3-4). According to their findings, individuals who endorse 'boundary national stories' are more inclined to vote for PRRPs. However, it is important to consider the possibility of overfitting if 'boundary national stories' are exclusively associated with an 'us versus them' narrative. If populist parties are the primary proponents of the 'us versus them' discourse, those who align with this idea are more likely to support populist parties.

3 – Still related to their research design, the authors use two questions to gauge "national stories." One is related to past events ("Which past event do you consider to be the most important?"), and the other is related to future events ("What future event do you wish to see?"). The authors claim that the responses to these open-ended questions can be used as proxies for, among others, national stories that are deeply rooted in "in-group favoritism" and "out-group hatred." However, this does not seem to be the case from the examples and results provided in the draft. There is no further explanation or even description of how answers such as "World War 2" can relate to this "us versus them" narrative/feeling.

4 – Theory-wise, the draft is based on concepts such as "populism" and "radical right," since it focuses on "populist radical right parties." However, it does not define either of these concepts.

5 – Data/method-wise, there are many inconsistencies along the draft. A few examples: variables present in some models but absent in others without explanation and variables presented during the text but not used in any of the models.

6 – Also, the authors utilize Laver, Gallagher, and Mair (2011) and the Parlgov to classify the parties as populist (PRRP) or not, but do not provide any further explanation. This becomes problematic when Parlgov does not explain how they classify a party as "populist" or "populist radical right." In this sense, a lack of transparency and reproducibility is concerning.

Reviewer #2: “Voices from the margins: How national narratives are linked to support for populist parties” is an exciting article that measures the impact of national narratives on voting for populist radical right parties or candidates. I thank the authors for their efforts. I present my comments, suggestions, and questions below, which I hope will help the authors (not in order of importance).

1) The authors refer to “national stories” that draw the borders of us versus them populist identities. However, if the authors talk about “national stories” or “national past,” then they focus on right-wing populist parties rather than left-wing ones. Therefore, there should be more discussion about nativism than populism. As Mudde (2007) argues, the populist radical right has three characteristics: nativism, populism, and authoritarianism. Also, it is nativism rather than populism that is the most prominent characteristic of these parties and candidates.

2) The authors did not explain the case selection. There should be at least a few sentences about why the authors examined Denmark, the Netherlands, the UK, and the US.

3) In the introduction (p. 4), the authors referred to the populist constituency as “the bottom of a society’s power structure.” On p. 12, they said that “marginalized people might cope with their subordinate position and at the same time strive to gain greater acceptance to mainstream

society by distancing themselves, physically or psychologically, from other marginalized

groups, and labeling the latter as outsiders and a threat to the community.” In short, the authors refer to the relative deprivation argument in the broadest sense rather than economic grievances. They also used varying sociodemographic variables in interaction models. Thus, it would be better to focus on relative deprivation literature more than losers of globalization. For additional literature, you can check societal pessimism (Steenvoorden & Harteveld, 2018), nostalgic deprivation (Gest et al., 2018), and collective nostalgia (Elçi, 2022).

4) For selecting populist parties, please also check the PopuList dataset (Rooduijn et al., 2023). Nevertheless, it would be better to acknowledge that voting for populist parties is not equal to having populist attitudes. There can be populists who vote for mainstream parties for mechanical and psychological reasons, especially in the UK.

5) The generation of national story variables is not clear. Is it a continuous scale or categorical? However, more importantly, what is the left-out group for stories? The authors used three of them in the same regression without having a multicollinearity issue. Please explain these issues in more detail.

6) Figures 1 a-d should be presented before explaining the results to provide a better picture to the readers. Also, average marginal effects rather than log odds could be reported to understand better and compare the magnitudes of independent variables.

7) The authors used different sociodemographic variables for interaction models. It is not easy to understand why they did not use the same ones in every country. Please explain how the socioeconomic variables were selected.

8) The authors did not explain the sampling and data collection processes. They should add a detailed explanation for each country either to the main text or the appendix.

8) Finally, as a robustness check, the authors could run the regressions without populist attitudes for the other three countries. In fact, they could also remove them from the analysis since they are not using a conventional populism battery.

References

Elçi, E. (2022). Politics of nostalgia and populism: Evidence from Turkey. British Journal of Political Science, 52(2), 697-714.

Gest, J., Reny, T., & Mayer, J. (2018). Roots of the radical right: Nostalgic deprivation in the United States and Britain. Comparative Political Studies, 51(13), 1694-1719.

Mudde, C. (2007). Populist radical right parties in Europe. Cambridge University Press.

Rooduijn, M., Pirro, A. L., Halikiopoulou, D., Froio, C., Van Kessel, S., De Lange, S. L., ... & Taggart, P. (2023). The PopuList: A database of populist, far-left, and far-right parties using expert-informed qualitative comparative classification (EiQCC). British Journal of Political Science, 1-10.

Steenvoorden, E., & Harteveld, E. (2018). The appeal of nostalgia: The influence of societal pessimism on support for populist radical right parties. West European Politics, 41(1), 28-52.

6. PLOS authors have the option to publish the peer review history of their article (what does this mean?). If published, this will include your full peer review and any attached files.

Reviewer #1: **Yes: **Eduardo Ryo Tamaki

Reviewer #2: No

---

## [Author Response · Author response to Decision Letter 0]

18 Nov 2023

Dear PLOS ONE Reviewers and Editorial Team,

Thank you for the invitation to revise and resubmit our paper, “Voices from the margins: How national narratives are linked with support for populist radical right parties” (Ms. No. PONE-D-23-23023). We thank the editor and the reviewers for their thoughtful and constructive comments, which have prompted us to significantly revise the paper. In the current version of the manuscript, we have expanded our discussion on national narratives and clarified how they differ from policy positions. We have also added a discussion on nativism and how it relates to national narratives. We have cross-referenced our classification of party-families with that of the PopuList dataset. We have revised the presentation of results, conducted several robustness checks, and bolstered confidence in our findings by excluding the populist attitudes as a control variable. 

Below we present each reviewer’s comments in italics, followed by a discussion of how we address them in the revised manuscript. We are grateful for your thoughtful engagement with our research and hope you will now find the paper appropriate for publication in PLOS ONE. 

Response to Reviewer 1

Reviewer 1 noted that "the paper presents an interesting and potentially novel idea." The reviewer added that s/he "liked the idea of testing the effect of ‘narratives’ (called here ‘national stories’) on the populist vote," and "thinks that there is much potential in this paper." The reviewer also raised some important questions and offered helpful suggestions for improving the work. We thank the reviewer for his/her support and constructive feedback, which allowed us to significantly improve the manuscript.

1.1 "Regarding their research design, the authors propose three research questions but can only answer (and hypothesize on) two."

Reviewer 1 is quite right, and we have revised the manuscript accordingly. In response to the comment, we have streamlined our research questions to two instead of three. The research questions are now as follows:

(a) What characterizes national stories held by populist radical right voters?

(b) What kind of voters are more affected by these stories?

We now better explain in the text how we address these two questions. To address the first question, we begin by providing descriptive association figures (Figure 1 a-d) illustrating the spatial proximity between stories and vote choice. The four graphs in Fig 1 represent the four countries in our sample and all highlight the separate positioning of mainstream and populist parties (or leaders). Next, through a series of regression models, we demonstrate that individuals embracing an “us versus them” Boundary narrative are more inclined to vote for populist radical right parties (PRRPs), whereas those embracing the other two types of narrative (Survival and Self-expression) are more likely to support mainstream parties.

Together, the aforementioned empirical analyses help provide an answer to our first research question. To address the second research question, we ran interaction models, which revealed that individuals we classify as “powerless” (i.e., low-educated, residing in rural areas, and with low social status) are more prone to translating the Boundary type of story into votes for populist parties compared to their more affluent counterparts.

1.2 In their research design and results, the authors argue that there are significant differences in the national stories embraced by voters of PRRPs compared to those of mainstream party supporters (introduction, pages 3-4). According to their findings, individuals who endorse 'boundary national stories' are more inclined to vote for PRRPs. However, it is important to consider the possibility of overfitting if 'boundary national stories' are exclusively associated with an 'us versus them' narrative. If populist parties are the primary proponents of the 'us versus them' discourse, those who align with this idea are more likely to support populist parties.

We appreciate the reviewer's comment, which prompted us to clarify the study’s theoretical foundations and the rationale for our primary dependent variable, the national story, in order to address concerns about potential overfitting or conflating national stories with policy agendas. We now elaborate in the paper that, while it is correct that populist radical right parties promote an “us versus them” rhetoric, they focus mainly on current politics. National stories, on the other hand, are least of all about current political affairs (see Table 1). They weave together the nation’s past historical events with future aspirations regarding it, and effectively constitute a blueprint for how people perceive their nation. Thus, theoretically, the construct of national story reflects a very broad perception of the nation – which, we contend, and empirically show, is a predictor of vote choice (p. 11). 

Furthermore, from an empirical standpoint, if Boundary national stories were linked exclusively to the agenda and rhetoric of populist radical right parties, we would expect a perfect alignment between holding such a story and voting for this kind of party. However, our analysis (Fig 3) demonstrates that this is not the case. Boundary national stories are also prevalent among voters for mainstream parties on both the right and the left side of the political spectrum.

1.3 Still related to their research design, the authors use two questions to gauge "national stories." One is related to past events ("Which past event do you consider to be the most important?"), and the other is related to future events ("What future event do you wish to see?"). The authors claim that the responses to these open-ended questions can be used as proxies for, among others, national stories that are deeply rooted in "in-group favoritism" and "out-group hatred." However, this does not seem to be the case from the examples and results provided in the draft. There is no further explanation or even description of how answers such as "World War 2" can relate to this "us versus them" narrative/feeling.

We thank the reviewer for prompting us to elaborate on our coding decisions and to clarify the ways we coded past and future story components. Following this comment, we now expand on our coding scheme, and coding classifications. On pages 17-18 we now write:

Boundary national stories establish a division between "the people," or the national community – “us” – and everyone outside it – “them.” These stories inherently position other nations as outgroups, emphasizing a separation between them and the ingroup (the nation). The “us-them” boundary is thus a site for conflict between the national community and the outgroups. According to our coding scheme, responses relating to conflicts and clashes between the nation and an outgroup were coded as Boundary. 

Survival stories, on the other hand, center around the nation’s economic and physical security and incorporate events and issues such as "the Industrial Revolution" and "Napoleonic wars" for the past, and "job opportunities" and "dealing with terrorism" for the future. Some events or issues lend themselves to a classification as both Survival and Boundary, and were accordingly relegated to both rubrics. For instance, narratives about wars or conflicts, such as "World War II," are salient to the nation’s physical security while at the same time marking its separation from other nations. Such elements were entered under both the Survival and the Boundary rubrics. 

Self-expression stories revolve around human rights, democracy and the quality of life with stories such as “women’s rights,” “peace,” "signing the Magna Carta," "suffrage," and "freedom" are coded as such. A detailed explanation of the coding implemented, along with intercoder reliability scores and descriptive statistics for the three story types, are provided in Appendix B.

1.4 Theory-wise, the draft is based on concepts such as "populism" and "radical right," since it focuses on "populist radical right parties." However, it does not define either of these concepts.

Thank you for bringing this to our attention. We now include a definition of radical right populist parties in the introduction:

This group of parties adhere to an ideology characterized by authoritarianism, nativism, and populism (Mudde 2007). They take radical positions on cultural issues with nativism, a combination of nationalism and xenophobia, being central to their ideology. They are regarded as populist, as they profess to represent the will of the people, as opposed to that of a putative “other,” commonly depicted as a corrupt and self-serving elite (Margalit 2019). 

Moreover, in Section 3 ("National stories and political behavior") we added a paragraph on nativism, a key feature in the ideology of populist radical right parties. 

A key feature of the PRRP ideology is nativism, a belief that "states should be inhabited exclusively by members of the native group (“the nation”) and that nonnative (or “alien”) elements, whether persons or ideas, are fundamentally threatening to the homogeneous nation state" (Mudde 2010: 1173 ). Indeed, the distinction between “us” and “them” is at the ideological core of populism and is touted by populist leaders, who shift all blame from “the innocent people” to either out-groups or “corrupt elites.” Notably, however, while advocating an” us vs. them” narrative, populist parties focus mainly on current politics. Stories, on the other hand, are least of all about current politics: They weave together the nation’s past events with future aspirations regarding it, and as such effectively constitute a blueprint for how people perceive their nation. Rooted in a very broad perception of the nation, a national story reflects the national identity of the group that embraces it. Accordingly, a national story does not deal with current policy and occurrences, which preoccupy political parties. 

1.5 Data/method-wise, there are many inconsistencies along the draft. A few examples: variables present in some models but absent in others without explanation and variables presented during the text but not used in any of the models.

Thank you for bringing this to our attention. We have ironed out inconsistencies throughout the paper. First, we reran our regressions and regenerated our coefficient plots. Now, all the control variables reported in the text (and only these variables) appear in our regressions and coefficient plots. Specifically, we write in the text:

The attitudinal and demographic variables we controlled for were derived from prior research on populist voting. The attitudinal covariates are populist attitudes, political knowledge, political interest, and ideological self-placement. The demographic variables are gender, age, education, social class, and a dummy variable for living in a rural region. The operationalization of all control variables, as well as their descriptive statistics, per country, can be found in Appendix D.

Consequently, we re-generated Fig 2 to include all control variables described above. Note that questions tapping populist attitudes are missing from the survey in the Netherlands, and thus the populist attitudes scale is not included in the figure on the Netherlands (the populist attitudes scale is also removed from the analysis as part of our robustness tests, see Fig F1 in the Appendix). 

Second, in the previous version of the text, to test H2, we interacted social demographic variables with the Boundary story and reported the results in Fig 4. Fig 4 includes only the socio-demographic variables for which we got substantive and statistically significant results for the effect of the interaction term (socio-demographic variable * Boundary story). Due to space limitation, we chose to display in Fig 4 only substantive and significant results on which we had reported. Following the reviewer's comment about inconsistencies, as well as a similar comment made by R2, we added to the Appendix the full results of this empirical exercise. The graph below includes interaction terms between all the socio-demographic attributes of powerlessness and the Boundary story (see Appendix in the revised version). No statistically significant results were obtained for the urban/rural divide in the UK; for social class in the Netherlands; or for education in the US and Denmark. 

 

1.6 Also, the authors utilize Laver, Gallagher, and Mair (2011) and the Parlgov to classify the parties as populist (PRRP) or not, but do not provide any further explanation. This becomes problematic when Parlgov does not explain how they classify a party as "populist" or "populist radical right." In this sense, a lack of transparency and reproducibility is concerning. 

Following the reviewer’s comment, we now explain in the text how we use these data sources to classify parties to party-families. Also, we cross-reference our categorization with the PopuList classification of radical right populist parties. There was a perfect match between our sources for classification of PRRPs and the PopuList’s. See here, for the case of Denmark; here for the classification of the PVV in the Netherlands as PRRPs, and here for the classification of the UKIP in the UK; The US is not included in the PopuList dataset. We added a footnote regarding this alignment of our classification to that of the PopuList:

Footnote # 2: Our classification of populist radical-right parties is also consistent with the PopuList classification (Rooduijn et al., 2023), which is based on country-specific experts’ and comparativists' assessment of these parties' core ideological attributes.

Response to Reviewer 2

Reviewer 2 stated that this was a “an exciting article that measures the impact of national narratives on voting for populist radical right parties or candidates.” At the same time, s/he offered several helpful suggestions for improving the manuscript. We are grateful for these insights and have worked diligently to implement the reviewer’s advice.

2.1 The authors refer to “national stories” that draw the borders of us versus them populist identities. However, if the authors talk about “national stories” or “national past,” then they focus on right-wing populist parties rather than left-wing ones. Therefore, there should be more discussion about nativism than populism. As Mudde (2007) argues, the populist radical right has three characteristics: nativism, populism, and authoritarianism. Also, it is nativism rather than populism that is the most prominent characteristic of these parties and candidates.

We thank the reviewer for pointing this out. We now clarify in the text that we focus on right-wing rather than left-wing populism. We thus also changed the title of the paper to include the words "radical right" in addition to "populist." We also added a discussion on the three core features of right-wing populism: nativism, populism, and authoritarianism; and as part of the discussion leading up to our first hypothesis, we state the expected relationship between nativism as an ideology and an “us versus them” Boundary story. We consequently cite Mudde 2007, 2011. On pages 10-11 we now write:

Our first theoretical expectation emanates from previous works on the importance of narratives, and specifically, on the idea that national stories reflect individuals’ political identities. If so, we expect to find differences between stories embraced by voters for mainstream and populist parties. Moreover, the demand side literature describes voters for PRRPs as having reverted to nativist values of exclusion and discrimination, in opposition to elites’ global and universalistic values (Mudde and Kaltwasser 2012), and as tending to glorify the past (Elçi 2022). Previous works claim that right-wing populist parties mobilize potential voters by fueling feelings of resentment and anger tapping on their outsiderness. These studies point at populist actors’ tendency to see the world in terms of us-versus-them dichotomies, be it “the people” versus the “corrupt elite” or negative sentiments toward out-groups. A key feature of the PRRP ideology is nativism, a belief that "states should be inhabited exclusively by members of the native group (“the natio

---

## [Decision Letter · Decision Letter 1]

23 Jan 2024

PONE-D-23-23023R1Voices from the margins: How national narratives are linked with support for populist radical right partiesPLOS ONE

Dear Dr. Oshri,

Thank you for submitting your manuscript to PLOS ONE. After careful consideration, we feel that it has merit but does not fully meet PLOS ONE’s publication criteria as it currently stands. Therefore, we invite you to submit a revised version of the manuscript that addresses the points raised during the review process. As R2 is satisfied with the revision, it appears that you accidentally omitted R1's previous review as it was an attachment to the email. You can now access both reviews provided by R1. I urge you to carefully read them and address each comment in detail. Also, given the comparative nature of your work, I recommend that you check out the related work published on populism in the field of political psychology.       

We look forward to receiving your revised manuscript.

Kind regards,

Cengiz Erisen

Academic Editor

PLOS ONE

Reviewers' comments:

Reviewer's Responses to Questions

**Comments to the Author**

1. If the authors have adequately addressed your comments raised in a previous round of review and you feel that this manuscript is now acceptable for publication, you may indicate that here to bypass the “Comments to the Author” section, enter your conflict of interest statement in the “Confidential to Editor” section, and submit your "Accept" recommendation.

Reviewer #1: (No Response)

Reviewer #2: All comments have been addressed

2. Is the manuscript technically sound, and do the data support the conclusions?

Reviewer #1: No

Reviewer #2: Yes

3. Has the statistical analysis been performed appropriately and rigorously? 

Reviewer #1: No

Reviewer #2: Yes

4. Have the authors made all data underlying the findings in their manuscript fully available?

Reviewer #1: Yes

Reviewer #2: (No Response)

5. Is the manuscript presented in an intelligible fashion and written in standard English?

Reviewer #1: Yes

Reviewer #2: Yes

6. Review Comments to the Author

Reviewer #1: I appreciate the effort and the changes made by the authors. Personally, I liked how the authors handled some of the comments regarding their research question, and I found that the changes improved heavily on the original manuscript. However, the third and fourth points remain problematic, and many (if not all) minor comments were left unaddressed. Since I believe this might be due to some problem that prevented them from accessing the file attached to the original review or maybe even an oversight, I recommend another round of revisions. I am attaching my original comments again.

Here are some other comments based on this round:

1) Related to the research design, I am still not convinced about how the questions used for both past and future events can be used as proxies for national stories “deeply rooted in in-group favoritism and out-group hatred.” Even though the authors addressed this comment in their response, I am still unconvinced.

a. “Boundary national stories establish a division between “the people,” or the national community – “us” and everyone outside it – “them.” These stories inherently position other nations as out-groups, emphasizing a separation between them and the in-group (the nation).” This type of story seems to be related to only nationalism. Both “national community” (treated as “us”) and “everyone outside it” (treated as “them”) seem to measure nationalism, as there is no reference to “elites” or the moral division between them and “the people,” or “us.” In this sense, if the authors are studying PRRPs (Populist Radical Right Parties), they will indeed find a relationship between “boundary national stories” and support for this party family. However, this connection will be driven mainly by nationalism and not populism.

b. How would “Napoleonic wars” and “Industrial revolution” (as past events) gauge a “survival story”? Given how long it has been since both events, can you confidently affirm that they still provoke sentiments related to economic and physical securities? I do not think this is the case.

c. How would “World War 2” mark the “separation from other nations”?

d. Finally, do the terms “survival” and “self-expression” come from studies in political culture? They seem to resemble Inglehart’s definitions heavily. If that is the case, why not acknowledge that?

2) There is still a problem with the concepts. My original comment was that the manuscript lacked a proper definition of concepts such as “populism” and “radical right” (consequently, “populist radical right”). The authors responded by adding a couple of phrases to the introduction and section 3. However, I do not think this is enough.

a. First, there is still no proper definition of populism.

b. Second, even though they added a couple of phrases in section 3, there needs to be a connection between “national stories” and “(radical right) populism.” What I mean by that is that the authors should properly connect these two things, developing how we should expect national stories to relate and connect to radical right populism.

c. Third, I would suggest caution when writing that “These studies point at populist actors’ tendency to see the world in terms of us-versus-them dichotomies, be it “the people” versus the “corrupt elite” or negative sentiments toward out-groups” (p. 11). Since populism requires anti-elitism, this sentence might be misleading.

d. Finally, I suggest splitting this paragraph (p. 11) into smaller parts.

3) When talking about the three types of stories, the authors state that “while the first two kind (survival and self-expression) encapsulate the norms, goals, and values of the national group, the Boundary story defines the national community relationally and negatively, in terms of exclusion.”

a. First, are the classifications of stories on “survival,” “self-expression,” and “us versus them boundary” mutually exclusive? Can boundary’s (us versus them) exclusion also be done through values and norms? Moreover, in this sense, would it still be “boundary” or something different?

b. Second, how can “survival” and “self-expression” expressed in stories relate to both material and economic survival (or freedom and emancipation) and “norms, goals, and values” at the same time? An example: how can “Napoleonic wars” relate to survival, in terms of material and economic security, and “norms, goals, and values of the national group” simultaneously? This seems to be stretching or reading in too much.

4) Have the authors tried using “stories” as moderators/mediators in their models? Although I am not entirely convinced yet of the importance of “stories” as predictors for support for Populist Radical Right parties, I think it makes sense to think of them as either moderators or mediators (depending on the authors’ theories and points of view).

5) Finally, I would suggest removing the variable “populist attitudes” from the models. First and most importantly, these variables (Appendix D) do not measure populist attitudes. While items 1 to 3 could be used, to some extent, items 4 to 8 measure different things, among which political efficacy (which has been commonly mistaken or even incorrectly used as a proxy for populist attitudes). Second, there is no further information on how the scale was created. If the authors plan on keeping these variables as a measure of populist attitudes, I would recommend (1) calling it something different, (2) displaying the result of an EFA or a CFA (depending on whether this scale was used before), and (3) explaining how the variables were aggregated, or how they come together to form a scale.

The rest of my initial comments are attached.

Reviewer #2: I congratulate the authors for putting enormous effort into replying to my comments and suggestions and revising the manuscript accordingly. I think the manuscript is ready for publication. I have one minor suggestion about Figure 2's presentation. While the note section indicates, "The boundary, survival, and self-expression are three scales for the three national story types," the graph is labeled as "us/them, survival, self-expression." It would be better to consistently use boundary or us/them throughout the manuscript.

7. PLOS authors have the option to publish the peer review history of their article (what does this mean?). If published, this will include your full peer review and any attached files.

Reviewer #1: No

Reviewer #2: No

---

## [Author Response · Author response to Decision Letter 1]

21 Mar 2024

Dear PLOS ONE Reviewers and Editorial Team,

Thank you for the invitation for a second round to revise and resubmit our paper, “Voices from the margins: How national stories are linked with support for populist radical right parties” (Ms. No. PONE-D-23-23023). We thank the reviewers for their feedback and specifically thank reviewer 1 and the editor for enabling us to work on a second round of revisions, as our previous handling of the comments was incomplete. In this version, we have shortened the introduction section for improved readability, conducted additional robustness checks, incorporated new literature from the social psychology field, and provided clearer explanations regarding the link between national stories and right-wing populism. Below we present the reviewers’ comments, followed by a discussion of how we address them in the revised manuscript. We apologize for the length of the memo, but we felt it was necessary in order to properly address all comments and suggestions. We will begin with addressing Reviewer 2's minor comments and then address the comments made by Reviewer 1. 

Response to Reviewer 2

2.1 The reviewer notes that they “congratulate the authors for putting enormous effort into replying to my comments and suggestions and revising the manuscript accordingly. I think the manuscript is ready for publication”. They have one minor suggestion regarding the presentation of Figure 2, recommending consistency in the labeling of the graph, particularly using "Boundary" instead of "us/them". 

Response: Point well taken. We fixed these inconsistencies in Figure 2. The renewed graph now uses the Boundary terminology, instead of us/them. 

Response to Reviewer 1

We are grateful for your thoughtful engagement with our research and hope you will now find the paper appropriate for publication in PLOS ONE. 

Reviewer 1 notes that they “appreciate the effort and the changes made by the authors.” They added that they “liked how the authors handled some of the comments regarding their research question, and I found that the changes improved heavily on the original manuscript.” However, they point out that the third and fourth points remain problematic, and many (if not all) minor comments were left unaddressed. 

We appreciate the valuable feedback provided by the reviewer and apologize for overlooking the minor comments, which were included in an attached document that we inadvertently did not open. We are grateful for the opportunity for a second round of revisions. Below, we address each comment and provide our responses accordingly.

1.1 The reviewer expresses skepticism regarding the use of questions about past and future events as proxies for national stories. Specifically, the reviewer argues that "boundary national stories," primarily measure nationalism rather than populism as it emphasizes the division between the national community (in-group) and everyone outside it (out-group), without considering the distinction between elites and the people. Consequently, the reviewer suggests that any relationship between "boundary national stories" and support for Populist Radical Right Parties (PRRPs) would likely be driven by nationalism rather than populism.

Response: We appreciate the reviewer's engagement with our research design and have taken their feedback into account in the revised manuscript. Firstly, we clarify that the national stories in our study, captured through past and future events respondents deem significant, primarily relate to nationalism rather than populist narratives. This distinction is now emphasized in the manuscript to provide clarity. Our aim is to analyze respondents' perceptions of their nation and the relationship of these perceptions with voting behavior, particularly in relation to support for populist radical right parties (PRRPs).

Secondly, we acknowledge the role of nativism in the populist ideology, a point raised in the previous round of review, and we now explicitly state in the text that the association between national stories and voting for PRRPs is driven, for the most part, by nativism rather than populism (see pages 3,6,11). We also suggest avenues for future research to explore populist narratives by aligning elements of populism, such as people centrism and anti-elitism, with the components of people's narratives (see page 31).

Furthermore, we acknowledge that the operational definition of national stories may not fully capture the breadth of national narratives and encourage future research to enrich the operationalization by incorporating additional elements such as peoples’ perception of the most important national character. This broader approach could lead to a deeper understanding of the narratives shaping political behavior (see page 31).

1.2 The reviewer questions the relevance of using historical events like the Napoleonic Wars and the Industrial Revolution as proxies for "survival stories" in the context of assessing contemporary sentiments related to economic and physical security. They express doubt about whether these events still evoke such sentiments given the considerable time elapsed since their occurrence.

Response: We appreciate the reviewer's in-depth consideration of the national story construct and its definition. In response to this comment, we have provided further elaboration on our classification logic in the revised manuscript (see pages 17-19). The classification of past and future events (story components) like the "Napoleonic Wars" and the "Industrial Revolution" as belonging to the "Survival" category is based on their inclusion of economic or conflict-related elements.

It is important to note that our classification pertains to the stories themselves, not the individuals holding them. We use a ‘text-oriented’ approach to reading the text rather than ‘author-oriented’ since we cannot determine the individual intentions of every respondent. While such reading of the text has its pros and cons, it makes sense to assume that if a person orients her national stories around survival-related content (e.g. economic or conflict-related elements), even historical ones, it suggests a priority or at least an attention to survival framing, in comparison with respondents who chose events relating to, say, women's rights. Additionally, we should keep in mind that past events are but one component of our proxy for national story, and a "full" survival story must also involve an event related to survival in the future.

We believe that future research could advance the understanding, definition, and measurement of national stories. Despite this limitation, our study aligns with previous empirical endeavors aimed at assessing people's national stories through surveys, which is made more explicit in this version. We acknowledge the trade-off inherent in our approach: while we benefit from a broad dataset encompassing a representative sample of respondents across different countries, our main independent variable, the national story construct, is defined and measured thinly. Our operational definition, using a sequence of past and future events to represent the national story, is but one approach, and we anticipate that future studies will refine and expand upon this strategy (see our discussion section). Nonetheless, our research stands as the first, to our knowledge, to establish a link between national stories and the vote for PRRPs, and we hope to see further exploration in this area.

1.3 The reviewer wonders how “world war 2” marks the separation from other nations.

Response: Following this comment our revised manuscript underscores that conflicts and wars inherently establish boundaries between at least two opposing groups. Therefore, story components containing WW2 as a past event include boundary elements of "us versus them."

1.4 The reviewer asks whether the terms “survival” and “self-expression” come from studies in political culture and if so, they suggest acknowledging Inglehart's work in the text.

Response: The reviewer is correct. We rely on Inglehart and colleagues' work, and specifically write the following:

These three categories mirror broad political divisions in advanced democracies. The first two are based on the work of Inglehart and Welzel [87, 88] and of Norris and Inglehart [7] in the World Value Survey, in which people’s value priorities are aggregated at the country level, and according to these aggregations, each country is then placed on a two-dimensional value continuum. On the first axis, self-expression values are at one pole and survival values at the other. Self-expression values emphasize autonomy, human and minority rights, freedom, environmental protection, and quality of life, whereas survival values center around economic and physical security [88-90]. 

1.5 The reviewer asks that we better define populism and radical right. They believe that further clarification is needed beyond the added phrases in the introduction and section 3.

Response: Following the reviewer comment the revised manuscript includes a definition of populist radical right parties, and populism. We specifically state that, similar to other (and many) empirical studies, we rely on the ideational approach to populism, that sees populism as thin-centered ideology that its core features include: anti-pluralism, anti-elitism and the juxtaposition of a virtuous people against elites. On page 15 we write the following:

Relying on the ideational approach to populism that understands politics as a Manichean struggle between the will of the homogenous people and the corrupt elite and which argues that political sovereignty should reside with the ordinary people [4, 18], we utilize the PopuList data to classify parties as populist and radical right parties

Importantly, our definition is consistent with that of the PopuList data. This change complements previous revisions we have implemented. In the introduction, we now state:

This group of parties adhere to an ideology characterized by authoritarianism, nativism, and populism [4] and take radical positions on cultural issues with nativism, a combination of nationalism and xenophobia, being central to their ideology. They are regarded as populist, as they profess to represent the will of the people, as opposed to that of a putative “other,” commonly depicted as a corrupt and self-serving elite [5].

1.6 The reviewer suggests that the connection between "national stories" and "(radical right) populism" needs further development. They recommend establishing a clearer connection between these concepts and explaining how national stories are expected to relate to radical right populism.

Response: We appreciate the reviewer's call for clarification of our theoretical premises and value the opportunity to enhance our text accordingly. In the updated version of the paper, we provide a clearer explanation of the connection between Boundary stories and the support for populist radical right. We specifically write the following: 

Our first theoretical expectation emanates from previous works on the importance of narratives, and specifically, on the idea that national stories reflect individuals’ political identities. If so, we expect to find differences between stories embraced by voters for mainstream and populist parties. Moreover, the demand side literature describes voters for PRRPs as having reverted to nativist values of exclusion and discrimination, in opposition to elites’ global and universalistic values [71], and as tending to glorify the past [66]. While populism and exclusionary nationalism, embodied in the Boundary type of national story, are analytically distinct constructs, they both play roles in the populist radical right. Populism revolves around the 'people as underdog' dynamic along an up-down axis, while exclusionary nationalism focuses on the 'people as nation' along an in-out axis. Despite their differences, both are integral components of the populist radical right. Previous works claim that right-wing populist parties mobilize potential voters by fueling feelings of resentment and anger, often leveraging "us-versus-them" dichotomies, whether between "the people" and the "corrupt elite" or through negative sentiments toward out-groups [72-74].

1.7 The reviewer advises caution regarding a statement suggesting that populist actors tend to perceive the world in terms of "us-versus-them" dichotomies, cautioning that this could be misleading given that populism requires anti-elitism.

Response: We thank the reviewer for this comment. We reworded this sentence, and now point to the strategy of PRRPs that often mobilize voters pitting them against the elites, or some others. We write the following: 

Previous works claim that right-wing populist parties mobilize potential voters by fueling feelings of resentment and anger, often leveraging "us-versus-them" dichotomies, whether between "the people" and the "corrupt elite" or through negative sentiments toward out-groups.

1.8 The reviewer suggests splitting the paragraph on page 11 into smaller parts.

Response: Point well taken. We splitted the paragraphs into smaller parts. 

1.9 The reviewer questions whether the classifications of stories as "survival," "self-expression," and "boundary" are mutually exclusive and whether the "boundary" type could also occur through values and norms. They inquire whether this would still fit within the "boundary" category or if it represents a different type of narrative.

Response: The reviewer's observation about our classification of story components into survival, self-expression, and boundary types is valid. These categories are not mutually exclusive; a story event can be classified under multiple types. For instance, World War II, a common past event, is classified as both survival and boundary. Importantly though, not all survival stories are boundary, and vice versa. This distinction is evident in our regression results, where survival tends to predict voting for mainstream parties, while boundary predicts voting for PRRPs.

Our classification is just one step in constructing the overall story. The second step involves combining past and future story components into a cohesive narrative that each individual holds. Respondents receive scores on the three indices of story types based on their composite narratives. For example, a respondent with a World War II past event and a peace future event would score 1 on all three indices, indicating elements of survival, boundary, and self-expression. Another respondent with a World War II past event and an economic prosperity future event would score 2 on survival, 1 on boundary, and 0 on self-expression.

1.10 The reviewer questions how story types such as "survival" and "self-expression" can encompass both material or economic concerns and the norms, goals, and values of a national group. 

Response: Consider a respondent who regards the "Napoleonic Wars" as the most significant past event and the "end of terror" as a significant future event. These events are classified as belonging to the survival category, placing the individual at the highest level on the survival scale. For us, this choice of events and not others, signifies that the respondent prioritizes the survival (security) of the nation over other goals, norms and values.

1.11 The reviewer suggests that we consider stories as moderators or mediators in our models, depending on our theories/point of view. 

Response: We thank the reviewer for their engagement with our argument and theoretical understanding on how stories may correlate with voter choice. We also truly think that more theoretical work is needed to connect stories with vote. In our manuscript, which to the best of our knowledge is the first to link stories with the vote for PRRPs, we link the two, with the moderating effect of voters’ marginal poisoning in society (low education, low status and the urban/rural divide), and show that boundary stories are indicative of the vote for PRRPs, only for those marginal groups in society. Those better off citizens who hold boundary stories, are less likely to vote for PRRPs. Following the reviewer comment we revised H2 (the hypothesis on the interaction effect) and accompanying text accordingly. We

---

## [Decision Letter · Decision Letter 2]

2 May 2024

PONE-D-23-23023R2Voices from the margins: How national stories are linked with support for populist radical right partiesPLOS ONE

Dear Dr. Oshri,

Thank you for submitting your manuscript to PLOS ONE. After careful consideration, we feel that it has merit but does not fully meet PLOS ONE’s publication criteria as it currently stands. Therefore, we invite you to submit a final round of revisions to the manuscript that address the points raised during the review process. The reviewer raises a number of important points that you will need to pay careful attention in the revisions. These comments are to-the-point and quite constructive, improving the manuscript's methodology and substantive contribution to the literature. While doing the revision, I also ask that you browse the literature in political science tackling far-right vote choice from different perspectives across Europe and the UK. Those would be particularly useful in extending the content of your work.  Please submit your revised manuscript by Jun 16 2024 11:59PM. If you will need more time than this to complete your revisions, please reply to this message or contact the journal office at plosone@plos.org. Please include the following items when submitting your revised manuscript:A rebuttal letter that responds to each point raised by the academic editor and reviewer(s). You should upload this letter as a separate file labeled 'Response to Reviewers'.A marked-up copy of your manuscript that highlights changes made to the original version. You should upload this as a separate file labeled 'Revised Manuscript with Track Changes'.An unmarked version of your revised paper without tracked changes. You should upload this as a separate file labeled 'Manuscript'.If applicable, we recommend that you deposit your laboratory protocols in protocols.io to enhance the reproducibility of your results. Protocols.io assigns your protocol its own identifier (DOI) so that it can be cited independently in the future. For instructions see: https://journals.plos.org/plosone/s/submission-guidelines#loc-laboratory-protocols. Additionally, PLOS ONE offers an option for publishing peer-reviewed Lab Protocol articles, which describe protocols hosted on protocols.io. Read more information on sharing protocols at https://plos.org/protocols?utm_medium=editorial-email&utm_source=authorletters&utm_campaign=protocols.

We look forward to receiving your revised manuscript.

Kind regards,

Cengiz Erisen

Academic Editor

PLOS ONE

Journal Requirements:

Reviewers' comments:

Reviewer's Responses to Questions

**Comments to the Author**

1. If the authors have adequately addressed your comments raised in a previous round of review and you feel that this manuscript is now acceptable for publication, you may indicate that here to bypass the “Comments to the Author” section, enter your conflict of interest statement in the “Confidential to Editor” section, and submit your "Accept" recommendation.

Reviewer #1: All comments have been addressed

2. Is the manuscript technically sound, and do the data support the conclusions?

Reviewer #1: Yes

3. Has the statistical analysis been performed appropriately and rigorously? 

Reviewer #1: Yes

4. Have the authors made all data underlying the findings in their manuscript fully available?

Reviewer #1: Yes

5. Is the manuscript presented in an intelligible fashion and written in standard English?

Reviewer #1: Yes

6. Review Comments to the Author

Reviewer #1: I would like to start by thanking the authors for the effort put into addressing each one of my previous comments. Not only were they meticulous in their responses, but they also covered all of the points I initially raised. I recognize the effort and time put into this, so I appreciate and congratulate them. With a few exceptions, I think all of my concerns were satisfactorily covered, and I am very satisfied with the answers provided by the authors. There are only a couple of things I would like to highlight (below). Other than that, I'm happy to recommend its publication.

First, some suggestions:

The authors' answer to comment 1.09 should be added to the paper. Even if as a footnote, an explanation of how the same event can have multiple categorizations (in the case of WW2, all three dimensions) is of great importance for the reader to properly understand the method used in the paper.

a. In the same note, if the example given by the authors, WW2 + Peace future event can receive a "1" for all three dimensions, would it then be counted as simultaneously all these three categories?

Comment 1.10: I believe that this was actually my mistake when writing the comment. What I meant is, how can a single story be related to both "survival" and "self-expression" if these two are diametrically opposed?

Comment 1.12: I'd like to see, if possible, a scree plot and a parallel test for the EFA the authors mentioned. In this case, if both tests point to alternatives with 1 or more factors, I'd like the authors to test the different alternatives. I suggest comparing them using chi-squared goodness-of-fit statistics and fit indices such as RMSEA, SRMR, and TLI. While I believe that it is possible for these items to load on the same dimension, I think it is important to highlight that Factor Analysis should always be used together with a solid theoretical base. In this case, even though the political efficacy question might load together with the other items (supposing the 1 factor solution is preferred over other alternatives), theory-wise, it is not populism.

Finally, two other comments that I think are relevant but have no impact on my final decision (as I explain below):

When answering point 1.2, the authors write:

“We appreciate the reviewer's in-depth consideration of the national story construct and its definition. In response to this comment, we have provided further elaboration on our classification logic in the revised manuscript (see pages 17-19). The classification of past and future events (story components) like the "Napoleonic Wars" and the "Industrial Revolution" as belonging to the "Survival" category is based on their inclusion of economic or conflict-related elements.

It is important to note that our classification pertains to the stories themselves, not the individuals holding them. We use a ‘text-oriented’ approach to reading the text rather than ‘author-oriented’ since we cannot determine the individual intentions of every respondent. While such reading of the text has its pros and cons, it makes sense to assume that if a person orients her national stories around survival-related content (e.g. economic or conflict-related elements), even historical ones, it suggests a priority or at least an attention to survival framing, in comparison with respondents who chose events relating to, say, women's rights. Additionally, we should keep in mind that past events are but one component of our proxy for national story, and a "full" survival story must also involve an event related to survival in the future.

We believe that future research could advance the understanding, definition, and measurement of national stories. Despite this limitation, our study aligns with previous empirical endeavors aimed at assessing people's national stories through surveys, which is made more explicit in this version. We acknowledge the trade-off inherent in our approach: while we benefit from a broad dataset encompassing a representative sample of respondents across different countries, our main independent variable, the national story construct, is defined and measured thinly. Our operational definition, using a sequence of past and future events to represent the national story, is but one approach, and we anticipate that future studies will refine and expand upon this strategy (see our discussion section). Nonetheless, our research stands as the first, to our knowledge, to establish a link between national stories and the vote for PRRPs, and we hope to see further exploration in this area.”

Although I appreciate their answer, I fear that I'm still not convinced. As far as I'm concerned, the authors have not convinced me that references to past (and even future) events such as the "Napoleonic Wars" and the "Industrial Revolution" can be classified as "survival" based on the "feelings of economic and physical securities" they elicit. The authors state that “while such reading of the text has its pros and cons, it makes sense to assume that if a person orients her national stories around survival-related content (e.g. economic or conflict-related elements), even historical ones, it suggests a priority or at least an attention to survival framing, in comparison with respondents who chose events relating to, say, women's rights." While I agree with the overall logic, I fail to grasp how these events represent an individual who "orients themselves around survival-related content." This explanation, to me, is one of the most important things missing in this paper.

Regarding comment 1.3, the authors write:

"Following this comment our revised manuscript underscores that conflicts and wars inherently establish boundaries between at least two opposing groups. Therefore, story components containing WW2 as a past event include boundary elements of “us versus them."

Again, I appreciate the authors' answer and the effort made in clarifying this point. However, I'm still not entirely sure about this. In the example I gave about WW2, I believe that the mere reference to "WW2" can elicit feelings that go beyond the boundaries of at least two opposing groups. Historical events of this magnitude can elicit not only "us versus them" feelings, but also many others related to things such as fear, loss, pride, shame, etc. Not all of them are necessarily tied to a "us versus them" divide. However, at this point, I believe these are limitations I personally have with the topic of the paper and not something that necessarily pertains to its quality or suitability for publication. While it does not influence my final decision, I think it's something worth mentioning.

7. PLOS authors have the option to publish the peer review history of their article (what does this mean?). If published, this will include your full peer review and any attached files.

Reviewer #1: No

---

## [Author Response · Author response to Decision Letter 2]

21 May 2024

Dear PLOS ONE editor and reviewer,

We thank the editor and the reviewer for their constructive comments. The revised version has benefited greatly from your constructive feedback and insightful suggestions. Below, we summarize our responses and revisions following the questions and suggestions that have been raised in your reports. 

Editor: the editor requested that we further engage with the literature on the support for far right parties in Europe and the UK. 

Response: We referenced Erisen & Vasilopoulou (2022), a comparative study of the Netherlands, the UK, and Germany, which highlights how citizens' emotional reactions, particularly anger, towards perceived immigration-related threats correlate with support for far-right parties. This citation adds to our review of empirical studies elucidating causal mechanisms behind far-right party support. We also reference Oesch (2008) who bring evidence from Austria, Belgium, France, Norway, and Switzerland for the support of manual workers in far-right parties. These studies integrate threat factors (economic or symbolic) with negative emotions or a declining trust. Following two rounds of revisions, we've broadened the paper's scope to encompass diverse literature on far-right party support bases. This includes insights from political psychology, comparative perspectives on voter choice, and political economy. Our bibliography now encompasses nearly 100 citations, reflecting the depth of our engagement with relevant scholarship.

Reviewer 1:

1. R1 suggests that our answer to comment 1.09 be added to the paper as it 

clarifies our coding logic further by, inter alia, explaining how the same event can have multiple categorizations. 

Response: We thank the reviewer for this suggestion and added this explanation as a footnote (#5). 

2. Relatedly, R1 wonders further about our example for a respondent holding ww2 past event and peace as a future event by asking if this respondent is simultaneously given the score of 1 in all three scales. 

Response: R1 is correct. The score is 1 in all three story-type categories on a scale of 0 to 2.

3. R1 clarifies his comment in 1.10. H/She was asking how can a single story be related to both "survival" and "self-expression" if these two are diametrically opposed?

Response: Indeed, story events cannot be classified as both survival and self-expression (they can however be classified as both survival and boundary). The reviewer is right that these two categorizations are diametrically opposed. 

4. R1 proposes distinguishing between the efficacy items and the populist attitudes items, which together form our populist attitudes scale in the previous draft. H/Se suggests conducting factor analysis or principal component analysis, followed by tests, to verify if these items are distinct not only theoretically but also empirically.

Response: We thank the reviewer for these suggestions. Following this comment, we rerun our analysis for the appendix (recall that we omitted from the main text the populist attitudes scale, as suggested previously by the reviewer). The populist attitudes scale was generated from eight items, each asks respondents to rate how much they agree or disagree with the statement. For convenience we list these statements here:

Statement 1: Political parties are interested in my vote, not in my opinion.

Statement 2: Politicians do not understand what is going on in society

Statement 3: There is a big gap between citizens and politics

Statement 4: It doesn't matter whom you vote for, the situation remains the same

Statement 5: Election campaigns give me enough information to make a choice

Statement 6: The political candidates argued in an open and sincere campaign

Statement 7: Parties can make a difference

Statement 8: I and my friends have an effect on government policy

For each country separately we conducted principal component analysis and subsequent tests, as suggested by the reviewer. Our analysis suggests (and this is in line with what the reviewer expected) that a two-dimensional summary (rather than 1) of the eight statements is very effective. As is clear from the left-hand plots, for the most part, statements 1-4 are loaded together in one factor, and the other statements are loaded in another. We consequently rerun our analysis in the Appendix, controlling for these two factors (and replaced this analysis with the one where we controlled for one factor aggregating the above 8 statements). Our results hold. 

 Note. Principal Component Analysis –Populist statements. Right panel: scree plots of the eight PCs and a parallel analysis for the eight statements. Left panel: Factor loadings for the eight statements in the first two PCs. 

The reviewer raised two interconnected comments that, we quote, "have no impact on my final decision". These comments are a follow-up to previous correspondence. The comments relate to our classification of stories to the three types of survival, self-expression and boundary. The reviewer writes that h/she is not fully convinced that "references to past (and even future) events such as the "Napoleonic Wars" and the "Industrial Revolution" can be classified as "survival" based on the "feelings of economic and physical securities" they elicit. H/she further writes that ""WW2" can elicit feelings that go beyond the boundaries of at least two opposing groups [and thus be classified as Boundary]. Historical events of this magnitude can elicit not only "us versus them" feelings, but also many others related to things such as fear, loss, pride, shame. 

Response:

To address the reviewer concern we first added a short paragraph (which is part of our previous answer to one of the reviewer's comments) in the text that clarifies to the reader our logic in classifying stories. We wrote in the text that "our classification pertains to the stories themselves, not the individuals holding them. We use a ‘text-oriented’ approach to reading the text rather than ‘author-oriented’ since we cannot determine the individual intentions of every respondent or the feelings these stories evoke. While such reading of the text has its pros and cons, it makes sense to assume that if a person orients her national stories around survival-related content (e.g. economic or conflict-related elements), even historical ones, it suggests a priority or at least an attention to survival framing of her national story, in comparison with respondents who chose events relating to, for example, women's rights." 

Second, we agree with the reviewer that people who prioritize a war as the most important past event of their nation can hold different national feelings beyond those delineated in this research (e.g., orienting the national stories around us versus), such as pride or fear. 

Given our research design which utilizes an open-ended survey question, we acknowledge the challenge of fully understanding the subjective meanings of stories to different individuals. Instead, our focus remains on analyzing the content of the responses, which has led us to categorize wars as historical events typically associated with us-versus-them or survivalist national frames. Having said that we do believe that future research can, and should, elaborate further this approach to account for possible interpretations and emotional effects of national stories, based on our typology or other typologies. Thus, we write in the discussion section, that we call for future research to refine our understanding and measurement of national stories. 

We end by thanking the reviewer again for the time and effort S/he put into his/her valuable suggestions. We believe the revised manuscript benefited tremendously from the constructive criticism.

---

## [Editor Report · Decision Letter 3]

2 Jun 2024

Voices from the margins: How national stories are linked with support for populist radical right parties

PONE-D-23-23023R3

Dear Dr. Oshri,

We’re pleased to inform you that your manuscript has been judged scientifically suitable for publication. Congratulations! Thank you very much for your fine contribution to PLOS ONE. I am pretty sure that when published, this paper will draw interest from the related literature. 

Kind regards,

Cengiz Erisen

Academic Editor

PLOS ONE
---

## [Editor Report · Acceptance letter]

3 Jul 2024

PONE-D-23-23023R3 

PLOS ONE

Dear Dr. Oshri, 

I'm pleased to inform you that your manuscript has been deemed suitable for publication in PLOS ONE. Congratulations! Your manuscript is now being handed over to our production team.

Kind regards, 

on behalf of

Dr. Cengiz Erisen 

Academic Editor

PLOS ONE